# Cholesterol efflux from C1QB-expressing macrophages is associated with resistance to chimeric antigen receptor T cell therapy in primary refractory diffuse large B cell lymphoma

Chimeric antigen receptor T (CAR-T) cell therapy has demonstrated promising efficacy in early trials for relapsed/refractory diffuse large B cell lymphoma (DLBCL). However, its efficacy in treating primary refractory DLBCL has not been comprehensively investigated, and the underlying resistance mechanisms remain unclear. Here, we report the outcomes of a phase I, open-label, single-arm clinical trial of relmacabtagene autoleucel (relma-cel), a CD19-targeted CAR-T cell product, with safety and efficacy as primary endpoints. Among the 12 enrolled patients, 8 experienced grade 4 hematologic toxicity of treatment-emergent adverse event. No grade ≥3 cytokine release syndrome or neurotoxicity occurred. Single-cell RNA sequencing revealed an increase proportion of *C1QB*-expressing macrophages in patients with progressive disease before CAR-T cell therapy. Cholesterol efflux from M2 macrophages was found to inhibit CAR-T cells cytotoxicity by inducing an immunosuppressive state in CD8$^+$ T cells, leading to their exhaustion. Possible interactions between macrophages and CD8$^+$ T cells, mediating lipid metabolism (*AFR1-FAS*), immune checkpoint activation, and T cell exhaustion (*LGALS9-HAVCR2*, *CD86-CTLA4*, and *NECTIN2-TIGIT*) were enhanced during disease progression. These findings suggest that cholesterol efflux from macrophages may trigger CD8$^+$ T cell exhaustion, providing a rationale for metabolic reprogramming to counteract CAR-T treatment failure. Chinadrugtrials.org.cn identifier: CTR20200376.

Diffuse large B cell lymphoma (DLBCL), a heterogeneous entity of B cell lymphoma that typically presents as an aggressive or advanced disease, represents 30–40% of all newly diagnosed non-Hodgkin lymphomas[1,2]. The combination of chemotherapy plus rituximab (CD20-targeting antibody) significantly improves the prognosis of patients with DLBCL[1,2]. However, approximately 20% of patients with primary refractory DLBCL[3] experience poor outcomes, with a median overall survival (OS) of only 7.1 months[4–6]. Several studies have demonstrated the efficacy and manageable adverse effects of chimeric antigen receptor T (CAR-T) cell therapy targeting CD19 in the treatment of relapsed/refractory (r/r) DLBCL[7–10]. In the ZUMA-7 and TRANSFORM trials, second-line CAR-T cell therapy was shown to be

✉e-mail: sls12280@rjh.com.cn; zhao.weili@yahoo.com

superior to standard-of-care chemotherapy with or without autologous stem cell transplantation (ASCT)[11,12], but less favorable survival was reported in primary refractory patients than in r/r DLBCL patients, highlighting the need for further improvement of efficacy in this population. Furthermore, approximately 60% of the patients in these trials experienced progressive disease (PD) following CAR-T cell therapy, with a median disease control period of 6 months[11,13-17]. Thus, understanding the mechanisms underlying resistance to CAR-T cell therapy in patients with primary refractory DLBCL is essential.

The tumor microenvironment (TME) plays essential roles in CAR-T cell therapy resistance[18,19]. The immunosuppressive nature of the TME promotes CAR T cell exhaustion and is a major obstacle to the clinical efficacy of the therapy[20]. Metabolic reprogramming of M2 macrophages has been reported to modulate the efficacy of immunotherapy and T cell function[21-23]. Indeed, cholesterol has been shown to induce immune checkpoint expression and $CD8^+$ T cell exhaustion within the TME[24]. Nevertheless, specific metabolic characteristics of TME, particularly those of M2 macrophages, in patients with DLBCL following CAR-T cell therapy remain of great interest. A phase I study of relmacabtagene autoleucel (relma-cel), a CD19-targeted CAR-T cell product with a 4-1BB costimulatory domain, confirmed the preliminary safety and efficacy of CAR-T cell therapy in r/r DLBCL[25]. This finding supported the rational design and initiation of a phase I, single-arm, open-label, multicenter clinical trial of relma-cel in primary refractory DLBCL (JWCAR029-003).

In the present study, we report the outcomes of this trial, including safety and efficacy of CAR-T cell therapy in patients with primary refractory DLBCL. In addition, we explore the mechanisms underlying resistance to CAR-T cell therapy in these patients using single-cell RNA sequencing (scRNA-seq).

## Results
### Safety and efficacy
Between 7th July 2020 and 21st June 2021, 19 patients were screened, and 14 patients who fulfilled the eligibility criteria were enrolled and subjected to leukapheresis (Fig. 1a). Relma-cel was successfully administered to 12 patients with a single infusion of $100 \times 10^6$ CAR-T cells. Notably, one patient (P002) was excluded from efficacy analysis due to a secondary-onset tumor (Hodgkin's lymphoma) at 13 months. The median time from enrollment to infusion was 34.5 days (range, 27–61 days). Patients' characteristics are shown in Supplementary Table 1. Four patients (33.3%) had sum of perpendicular diameters (SPDs) of ≥5000 mm² before CAR-T cell therapy, and five patients (41.7%) received bridging therapy with second-line chemotherapy: three patients received the ICE regimen (etoposide, ifosfamide, and carboplatin), one received R-ICE (rituximab, etoposide, ifosfamide, and carboplatin), and one received the combination of dexamethasone, cyclophosphamide, and vindesine.

As shown in Supplementary Table 2 at least one treatment-emergent adverse event (TEAE) of any grade occurred in all 12 patients. Of these 12 patients, 8 (66.7%) experienced a TEAE of grade 4. However, all grade 4 TEAEs, including leukopenia ($N = 3$), lymphopenia ($N = 5$), neutropenia ($N = 5$), and thrombocytopenia ($N = 2$), were related to investigation. Cytokine release syndrome (CRS) was observed in 6 of the 12 patients (50.0%; 4 patients experienced grade 1 CRS and 2 patients experienced grade 2 CRS). The median time of CRS onset was 5 days (range, 0–7 days), with a median duration of 10 days (range, 1–17 days). Neurotoxicity (NT), including muscle spasm and somnolence, was observed in two patients (16.7%, grade 1 with reversible symptoms). The median time of NT onset was 15 days (8 and 22 days, respectively), and the median duration was 5 days (7 and 3 days, respectively). For CRS management, tocilizumab (8 mg/kg) was administered in three (25.0%) patients, and glucocorticoids were not prescribed. One patient (P006) died 10 months after infusion due to renal abscess resulting from *Listeria* infection, which was further complicated by *Pseudomonas aeruginosa* septicemia. The death was considered unrelated to CAR-T cell therapy.

Eleven patients underwent efficacy assessment (Fig. 1b). At 1 month, four patients achieved a complete response (CR) and four patients achieved a partial response (PR), with 1-month CR rate (CRR) and overall response rate (ORR) of 36.4% and 72.8%, respectively. At 3 months, five patients had PD, with 3-month CRR and ORR of 36.4% and 45.5%, respectively (Fig. 1c). Post-treatment imaging evaluations showed that the median best percentage change of SPD from baseline was −64.7% (range, −97.43 to −54.56%), with six (54.5%) patients having a >50% decrease from baseline, four of whom achieved CR (Fig. 1d). With a median follow-up time of 10.0 months (range, 1.2–24.0 months), the median progression-free survival (PFS) and OS were 1.5 (95% confidence interval [CI], 1.0 to NE) months and 4.5 (95% CI, 3.0 to NE) months, respectively, for patients with SD (stable disease)/PD, while they were not reached in CR/PR patients (Fig. 1e). The results of other secondary endpoints of patients are shown in Supplementary Table 3.

### Cellular kinetics and serum biomarkers
The expansion curve of CAR-T cells in the 11 patients whose pharmacokinetics (PK) was evaluable is shown in Supplementary Fig. 1a. No significant difference in the peak value of CAR-T cells was found between CR/PR and SD/PD patients at any timepoint of expansion from day 1 to day 365 after infusion (Supplementary Fig. 1b); moreover, no significant differences in the peak value, peak day, and $AUC_{1-29}$ were observed (Supplementary Fig. 1c). Surprisingly, the ex vivo potency of cytotoxicity of CAR-T cells in CR/PR patients was significantly lower than that in SD/PD patients ($p = 0.0295$, Supplementary Fig. 1d). Serum biomarker data of CR/PR and SD/PD patients as well as those with or without CRS/NT are presented in Supplementary Fig. 2a, b. In brief, the levels of IL-6 and IL-1β were significantly elevated in patients with CRS grade 2 compared with those in patients with CRS grades 0 and 1 (Supplementary Fig. 2c). The IL-6 level was also increased in patients with NT grade 1 compared with that in patients with NT grade 0 (Supplementary Fig. 2d).

### TME features before CAR-T cell therapy
To obtain a deeper insight into the effect of TME on CAR-T cell therapy, fresh tissue samples were collected via core needle biopsy on day −6 (1 day before lymphodepletion) (before CAR-T cell therapy), on day 9 after relma-cel infusion when adequate CAR-T cell expansion was detected (during CAR-T cell expansion), and at 3 months after relma-cel infusion if the disease progressed (during disease progression). We performed scRNA-seq longitudinally on 10 fresh biopsy tissues from 5 patients with primary refractory DLBCL, including 5 samples before CAR-T cell therapy (CR, $N = 2$; PD, $N = 3$), 3 samples during CAR-T cell expansion, and two samples during disease progression; 2 samples were collected from the same 2 patients across the three timepoints (Fig. 2a).

Based on gene expression profiling, isolated cells from the biopsy tissues were categorized into five primary clusters, namely, B cells, T/NK cells, myeloid cells, endothelial cells, and fibroblasts (Supplementary Fig. 3a). Uniform Manifold Approximation and Projection (UMAP) demonstrated the distribution of these five clusters among different samples, treatment responses, and timepoints (Fig. 2b). To elucidate their associations in response to CAR-T cell therapy, myeloid and T cells were further divided into subclusters in tumor samples before CAR-T cell therapy. Myeloid cells were distributed over seven subclusters, namely, cDC1, cDC2, cDC3, pDC, IL1B macrophages, C1QB macrophages, and low-quality cells, based on gene expression profiles (Fig. 2c). In detail, the cDC1 subcluster specifically expressed *CLEC10A* and *CD1C*; the cDC2 subcluster expressed *CLEC9A* and *XCR1*; the cDC3 cluster expressed *LAMP3* and *IDO1*; the pDC subcluster expressed *CLEC4C*, *TCF4*, and *IRF7*; IL1B macrophages expressed *IL1B* and *FCN1*; and C1QB macrophages expressed *C1QB, APOE*, and *CD163*, which are main phenotypic markers for M2 macrophages (Supplementary

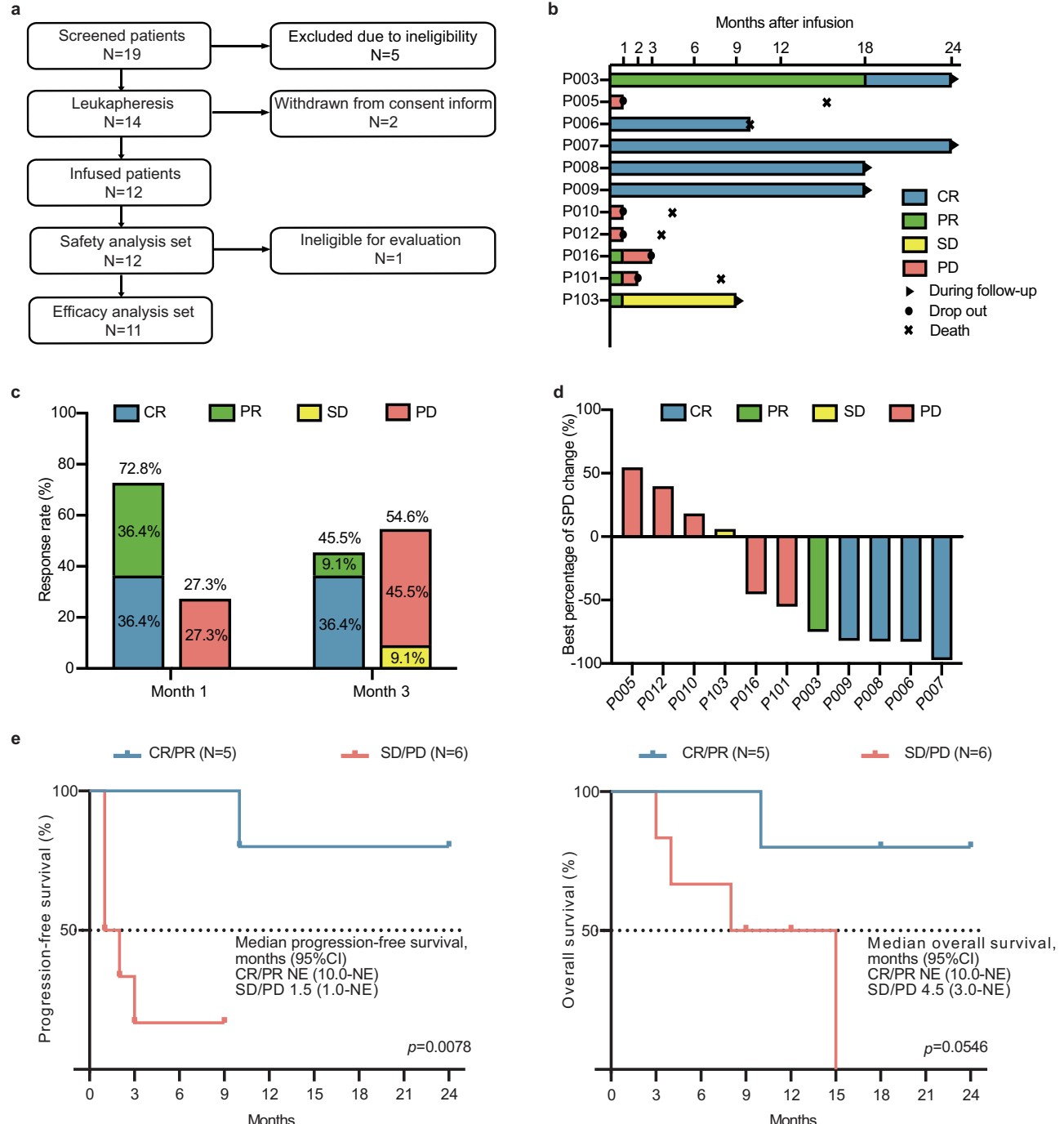

**Fig. 1 | Efficacy of CAR-T cell therapy in patients with primary refractory DLBCL.** **a** Participant flowchart. **b** Records of the time of treatment response change in 11 patients who received CAR-T cell therapy. **c** Treatment response following 1 month (day 29) and 3 months (day 90) of CAR-T cell infusion. **d** The best percentage of SPD change from baseline. **e** Progression-free survival and overall survival in patients with objective response (CR/PR, $n = 5$) and no response (SD/PD, $n = 6$). Statistical analysis was performed using log-rank tests. CR complete response, PR partial response, SD stable disease, PD progressive disease, SPD sum of perpendicular diameters.

Figs. 3b, c and 4). The proportion of C1QB macrophages, the most common subcluster, was increased in patients with PD compared with that in patients with CR before CAR-T cell therapy (Fig. 2d). T cells were further categorized into 16 subclusters, including 10 CD8+ subclusters, 4 CD4+ subclusters, 1 NK subcluster, and 1 low-quality subcluster (Fig. 2e). The NK subcluster expressed high levels of *GNLY* and *KLRB1*, whereas the Teff subcluster expressed *FGFBP2*, *GZMH*, and *GZMB*. Checkpoint genes (*LAG3*, *HAVCR*, *TIGIT*, and *CTLA4*) were expressed in CD8 Tex and CD8 Prolif cells, among which Tex-2, -3, -4, and

-6 subclusters commonly expressed *CXCL13*. CD8 Prolif expressed *CXCL13* and the cell cycle-related genes *MKI67* and *TOP2A*. The CD4 naive subcluster expressed *SELL* and *IL7R*, whereas CD4 Treg expressed *FOXP3*; CD4 mem (memory) expressed *IL7R* and *ANXA1*; and CD4 Th1-like expressed *TIGIT*, *CTLA4*, and *TNFRSF4* (Supplementary Fig. 3d, e). As shown in Fig. 2f, there was no significant difference in T cells between patients with CR and those with PD (the percentages of CD8 Tex cells in patients with CR vs. those with PD: 56.25% vs. 48.33%; CD8 Teff cells: 4.20% vs. 4.76%; and CD8 Prolif cells: 4.98% vs. 5.35%).

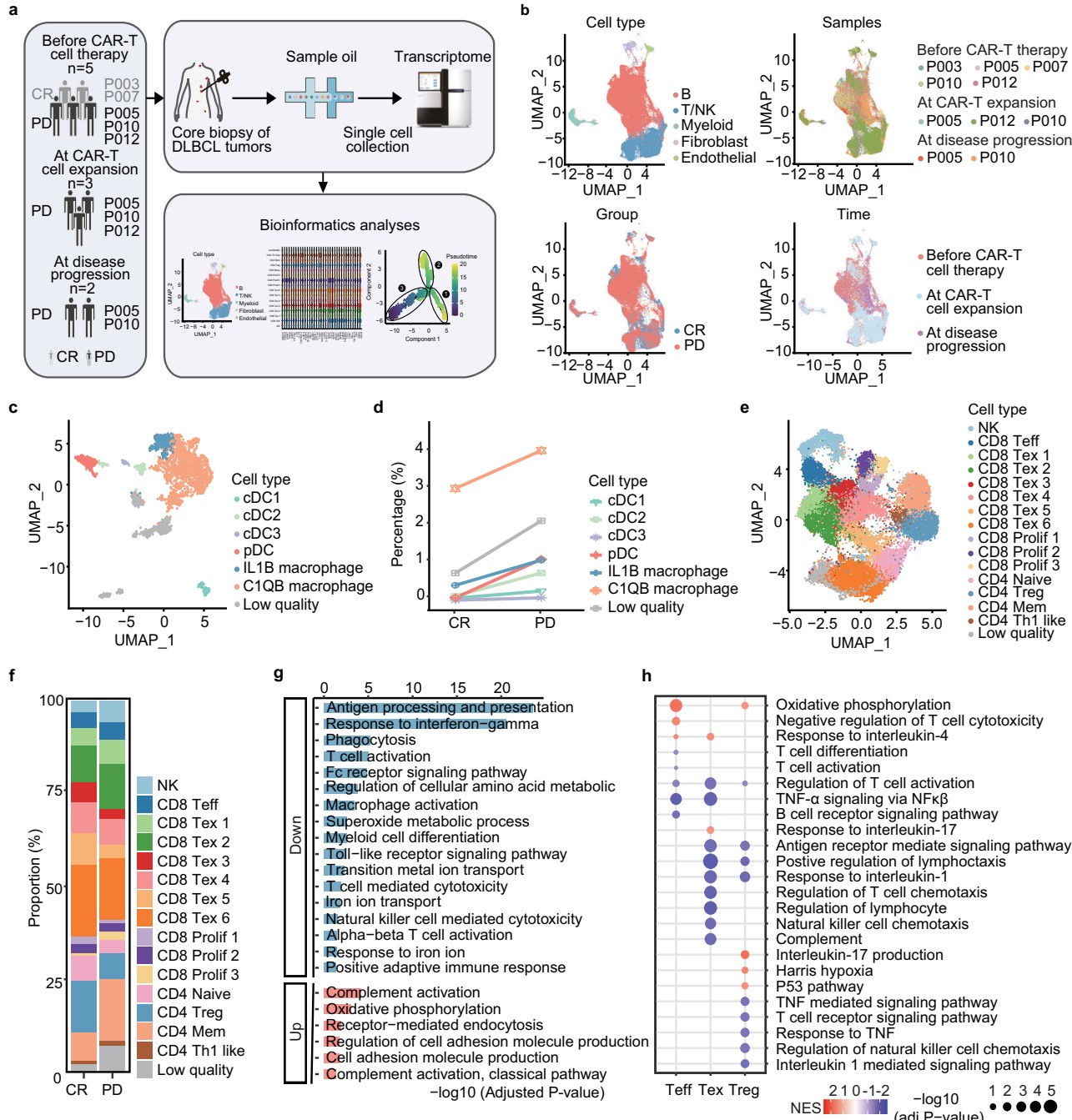

**Fig. 2 | Tumor microenvironment characteristics in patients with CR and PD before CAR-T cell therapy. a** Flowchart of scRNA-seq. **b** Distributions of five common cell clusters among different samples, treatment responses, and time-points. **c** Myeloid cells were divided into seven subclusters: cDC1, cDC2, cDC3, pDC, IL1B macrophages, C1QB macrophages, and low-quality cells. **d** Percentages of the cDC1, cDC2, cDC3, pDC, IL1B macrophage, C1QB macrophage, and low-quality subclusters in patients with CR and PD before CAR-T cell therapy. **e** T cells were further divided and a total of 16 subclusters were identified, including 10 CD8[+], 4 CD4[+], 1 NK, and 1 low-quality subclusters. **f** Percentages of the 16 subclusters of

T cells in patients with CR and PD before CAR-T cell therapy. **g** Enriched signaling of the genes related to C1QB macrophages identified from patients with CR and PD before CAR-T cell therapy. The adjusted *p* value was calculated using GO by default hypergeometric test. One-sided *p* value was calculated. **h** Enriched signaling of differentially expressed genes (PD vs. CR) identified in Teff, Tex, and Treg populations. NES normalized enrichment score. The adjusted *p* value was calculated using GSEA. Two-sided *p* value was calculated. *n* = 2 for CR group and *n* = 3 for PD group before CAR-T cell therapy.

Based on the pathways enriched for differentially expressed genes (DEGs), C1QB macrophages in patients with CR showed high expression of genes related to immune activation signaling, including antigen processing and presentation, response to interferon-gamma and T cell activation, whereas patients with PD showed high expression of genes related to immunosuppression signaling, including oxidative

phosphorylation, complement activation, and receptor-mediated endocytosis (Fig. 2g). Meanwhile, patients with PD showed low expression of genes related to immune activation signaling in Teff cells, such as T cell activation and TNF-α signaling, and low expression of genes related to immune activation signaling in Tex cells, such as antigen receptor-mediated signaling pathway and T cell chemotaxis

regulation (Fig. 2h). This may have been due to the relatively small sample size and panel size in our study. These results indicated that C1QB macrophages were in an activated state, whereas T cells were in an immunosuppressive state in patients with PD before CAR-T cell therapy.

## Increased cholesterol efflux from C1QB macrophages in patients with PD

Utilizing the scRNA-seq data of patients with PD, we longitudinally assessed changes in C1QB macrophages before CAR-T cell therapy, during CAR-T cell expansion, and during disease progression, and the results showed that the percentage of C1QB macrophages gradually increased during CAR-T cell therapy (4.17%, 5.96%, and 25.59%, respectively, Fig. 3a). DEGs identified between CAR-T cell expansion/ disease progression and before CAR-T cell therapy were mainly enriched in immunosuppression-related pathways, such as cholesterol homeostasis, cholesterol efflux, positive regulation of cholesterol esterification, positive regulation of cholesterol efflux, and lipid catabolic process (Fig. 3b). Since C1QB macrophages expressed main phenotypic markers for M2 macrophages, we applied M2 macrophage to further explore the effect of macrophage cholesterol efflux on CAR-T cell function. To examine the effect of the TME of DLBCL cells on M2-polarized macrophages' cytomembrane cholesterol levels, control cells, M2 and M2 ABCA1-knockdown (membrane cholesterol efflux transporters) macrophages (M2$^{shABCA1}$) were cultured with conditioned medium from DLBCL (DB) cells (Fig. 3c). Cholera toxin subunit B (CTB) staining, which is commonly used to analyze cholesterol-rich membrane microdomains, revealed reduced cytomembrane cholesterol levels in M2 macrophages compared with those in control cells; however, this was rescued by ABCA1 knockdown (Fig. 3d). Furthermore, in our coculture model established to examine the effects of M2 macrophages on CAR-T cell cytotoxicity against CD19-expressing DLBCL cells (DB cells) and the impact of CAR19 cells on M2 macrophages' total cholesterol levels either within or secreted by M2 macrophages (Fig. 3e), M2 macrophages exhibited significantly lower cholesterol levels than control cells in the presence of CAR19 cells and a higher cholesterol level in the coculture medium. However, this was restored by ABCA1 knockdown (Fig. 3f). Moreover, in the presence of M2 macrophages, the cytotoxic effect of CAR19 cells on DB cells, but not CD19-negative acute promyelocytic leukemia cells (HL60 cells), was markedly inhibited compared with that in the control cells; however, it was restored by ABCA1 knockdown (Fig. 3g). To validate the impact of cholesterol efflux on the immunosuppressive properties of M2 macrophages, we utilized 9-cis-retinoic acid (9cRA), pharmacological approach which was previously used to promote cholesterol efflux in M2 macrophages (Supplementary Fig. 5a)[26]. As expected, a significant decrease in total cholesterol content in the co-culture system was observed after 9cRA treatment, accompanied by an increase in cholesterol content in the culture medium (Supplementary Fig. 5b). Notably, macrophages exhibited significant polarization toward the M2 phenotype as assessed by CD206 expression (Supplementary Figs. 5c and 8a). In CAR-T cell cytotoxicity experiments, it was observed that the cytotoxicity of CAR-T cells in the M2$^{9cRA}$-CAR19 coculture group was significantly decreased when compared to control group (Supplementary Fig. 5d). These findings collectively suggest that the increased cholesterol efflux in M2 macrophages inhibits the anti-tumor function of CAR-T cells.

Furthermore, M2 macrophages significantly decreased the degranulation responses of CD4$^+$/CD8$^+$ CAR-T cells, which were also rescued by ABCA1 knockdown (Supplementary Figs. 6a–d and 8b). The percentage of effector memory T (Tem), terminally differentiated effector memory T (Temra) cells was increased, whereas the percentages of naive, central memory T cells (Tcm) cells (Supplementary Fig. 6e–g), and PD1$^-$LAG3$^-$TIM3$^-$ cells (Supplementary Figs. 6h, i and 8c) were decreased in the group with the coculture of CAR-19 and M2 macrophages; however, these were restored by ABCA1 knockdown.

These results indicated that increased cholesterol efflux from M2 macrophages in patients with PD impaired the cytotoxicity of CAR-T cells against tumor cells.

## CD8$^+$ T cell exhaustion during disease progression in patients with PD

Based on the previously reported dysfunctional and immunosuppressive signatures[27–30], 10 T cell subclusters were scored. Most of the CD8 Teff and CD8 Tex cells had higher dysfunctional signature and immunosuppressive signature scores in patients with PD (Fig. 4a). In particular, signature scores of both CD8 Teff and CD8 Tex 1–6 subclusters were significantly increased during CAR-T cell expansion and/or disease progression compared with those before CAR-T cell therapy in patients with PD (Fig. 4b). Genes related to Teff and Tex cells involved in oxidative phosphorylation, glycolysis, and cellular response to hypoxia signaling were significantly upregulated during disease progression compared to pre-CAR-T cell therapy or CAR-T cell expansion. In contrast, genes involved in TNFα signaling via NFκB in CD8 Teff and CD8 Tex cells, as well as those involved in T cell activation related to CD8 Teff cells, were significantly downregulated during disease progression compared with those before CAR-T cell therapy or during CAR-T cell expansion (Fig. 4c). This indicates an immunosuppressive state of T cells during disease progression. Three main states of CD8$^+$ T cells were characterized via trajectory analysis (Fig. 4d): state 1 involved an equilibrium between activated Teff and immunosuppressive Tex and Tex-Prolif cells; state 2 exhibited a high proportion of Tex cells; and state 3 exhibited a lower proportion of Tex cells than state 2 and a high proportion of Tex-Prolif cells with exhaustion characteristics. According to the distribution characteristics of each state at different timepoints, CD8$^+$ T cells were mainly distributed in state 1 before CAR-T cell therapy, in state 2 during CAR-T cell expansion, and in state 3 during disease progression (Fig. 4e), suggesting that CD8$^+$ T cells were gradually exhausted during CAR-T cell therapy in patients with PD.

Furthermore, in Teff cells, the expression of genes related to effector molecules (such as IFNGR1, KLRG1, KLRB1, and GZMK) and transcription factors (such as KlF13, HMGN1, KLF12, and TFDP2) was gradually decreased following CAR-T cell expansion and disease progression, whereas in Tex cells, genes related to inhibitory molecules (CCL3, TIGIT, CTLA4, and PDCD1) and transcription factors (PRDM1, IRF7, IRF9, and TOX2) were upregulated during CAR-T cell expansion and disease progression compared with those before CAR-T cell therapy (Fig. 4f). These findings suggest that T cells were in an exhausted state during disease progression in patients with PD.

## CD8$^+$ T cell exhaustion triggered by ligand-receptor interactions between C1QB macrophages and CD8$^+$ T cells

Using the Cellchat software to assess cell communication between C1QB macrophages and CD8$^+$ T cells during CAR-T cell therapy, extensive cell-cell communication between C1QB macrophages and CD8$^+$ T cell subclusters was observed during CAR-T cell expansion and disease progression (Fig. 5a), including that between C1QB macrophages and immunosuppression- or exhaustion-associated ligand receptors (such as ARF1-FAS, CD86-CTLA4, NECTIN2-TIGIT, and HBEGF-CD44) (Fig. 5b). Consistent with this, APOE-SORL1 communication only occurred before CAR-T cell therapy, whereas ARF1-FAS communication occurred during CAR-T cell expansion and disease progression (Fig. 5b). This finding suggested that C1QB macrophage-expressed ARF1 may induce apoptosis of CD8 Teff, CD8 Tex, and CD8 Prolif cells during CAR-T cell expansion and disease progression. Furthermore, we examined the expression of C1QB macrophages (APOE, ARF1, LGALS9, NECTIN2, and CD86) and CD8 Teff, CD8 Tex, and CD8 Prolif cell receptors (SORL1, FAS, HAVCA2, TIGIT, and CTLA4) at three timepoints in patients with PD. SORL1, the lipid metabolism-related receptor, was expressed in CD8$^+$ T cells before CAR-T cell therapy,

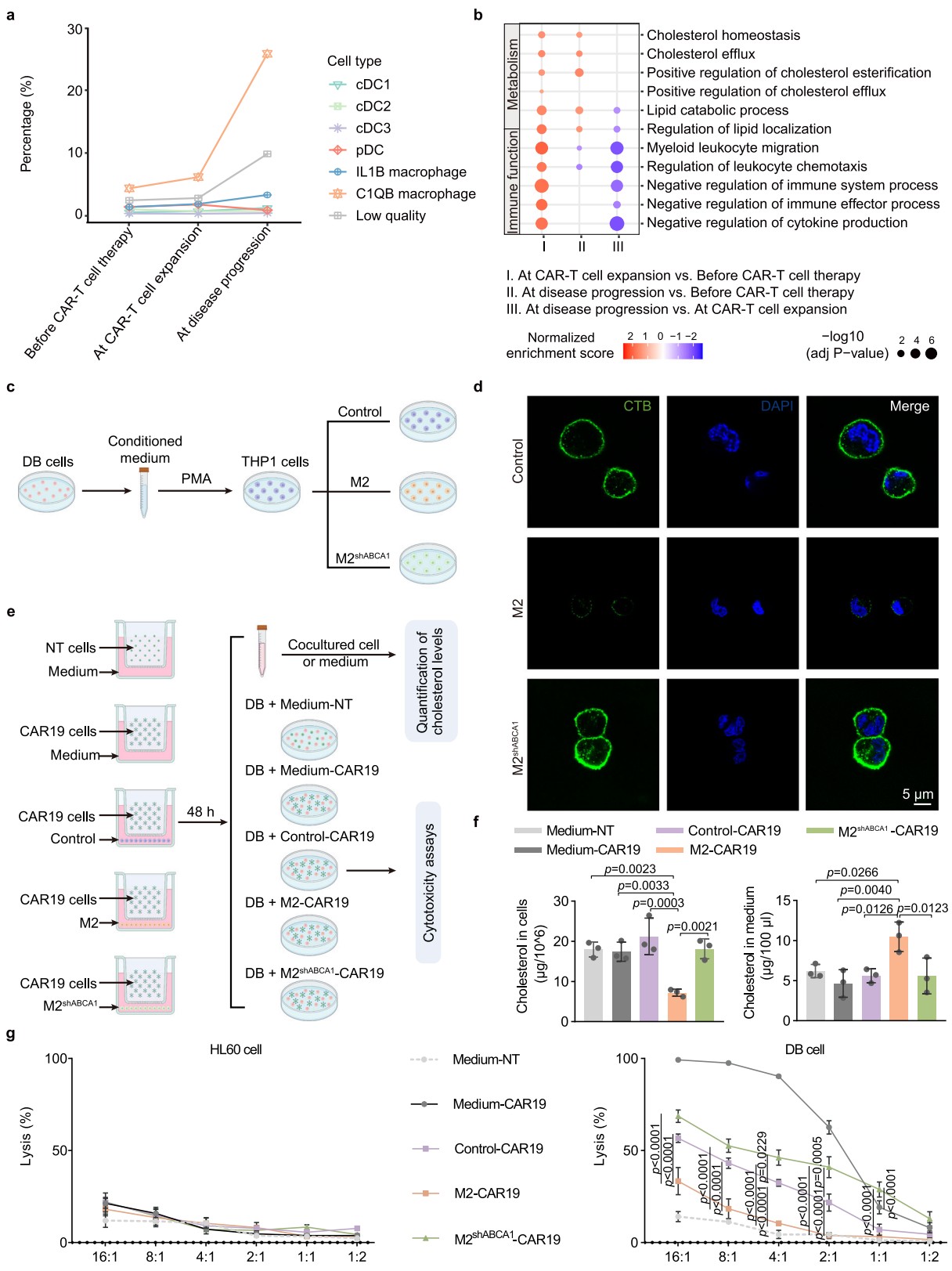

**f** (figure legend)
Medium-NT | Control-CAR19 | M2^shABCA1-CAR19
Medium-CAR19 | M2-CAR19

**g** legend:
Medium-NT
Medium-CAR19
Control-CAR19
M2-CAR19
M2^shABCA1-CAR19

whereas HAVCR2 and CTLA4, the immune checkpoints/T-exhaustion-related receptors, only appeared during CAR-T cell expansion and disease progression. Although TIGIT, another immune checkpoint-related CD8+ T cell receptor, was expressed during CAR-T cell therapy, its ligand, NECTIN2, was only detected before this therapy (Fig. 5c). These results indicated that the crosstalk between C1QB macrophages and CD8+ T cells depended on APOE-SORL1, NECTIN2-TIGIT, LGALS9-

HAVCR2, and CD86-CTLA4 communication, thus contributing to T cell exhaustion during CAR T expansion and disease progression.

## Cholesterol efflux from C1QB macrophages induced T cell exhaustion through TCR signaling inactivation

Cholesterol accumulation in the TME can induce CD8+ T cell exhaustion[24]. Here, we verified the effect of cholesterol efflux from

**Fig. 3 | Increased cholesterol efflux from M2 macrophages in patients with PD.** **a** Percentages of M2 macrophages in patients with PD before CAR-T cell therapy, during CAR-T cell expansion, and during disease progression. **b** Gene Set Enrichment Analysis for M2 macrophages of patients with PD at different timepoints. **c** In vitro, the culture supernatants collected from DB cells were used to culture control cells, M2 cells, and ABCA1-knockdown M2 (M2^shABCA1 cells). **d** Cytomembrane cholesterol levels in M2 macrophages were assessed via immunofluorescence using the Vybrant Alexa Fluor 488 lipid raft labeling kit. **e** Coculture models were constructed to assess the effects of M2 macrophages on CAR-T cell cytotoxicity against CD19-expressing DLBCL cells (DB cells) and the impact of CAR19 cells on M2

macrophages' total cholesterol levels either within cells or secreted into the coculture medium. **f** Total cholesterol levels in the cocultured M2 macrophages and cell-free cocultured media were measured in the above indicated groups. **g** Cytotoxic effect of CAR19 cells on CD19-expressing DLBCL cells (DB) and CD19-nonexpressing acute promyelocytic leukemia cells (HL60) was detected using a luciferase-based CTL assay. Data are shown as mean ± s.e.m. Statistical analysis was performed using two-way ANOVA with Tukey's multiple comparison tests. $n = 3$ for before CAR-T cell therapy group, $n = 3$ for at CAR-T cell expansion group, and $n = 2$ for at disease progression group in patients with PD.

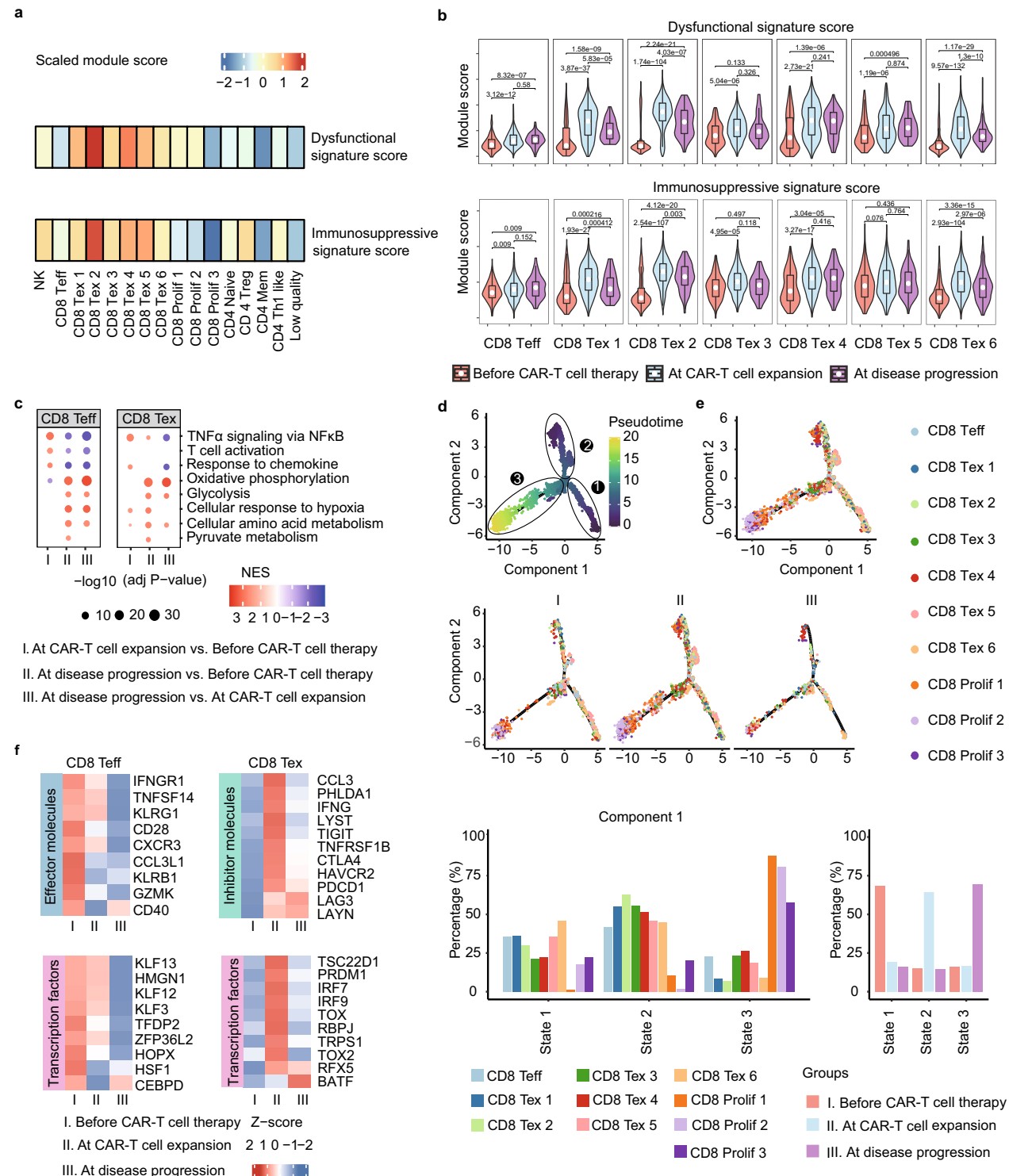

**Fig. 4 | Exhausted CD8+ T cells during disease progression in patients with PD.** **a** T cells in 16 subclusters were scored according to the dysfunctional signature and immunosuppressive signature. **b** The dysfunctional signature and immunosuppressive signature scores of CD8 Teff and CD8 Tex 1-6 subclusters were assessed in patients with PD before CAR-T cell therapy, during CAR-T cell expansion, and during disease progression. *n* = 8374 CD8 T cells were used for visualization, excluding those with a score of 0 due to the absence of detected gene expression. In the boxplot, a white dot within the box marks the median. The bottom and top of the box are located at the 25th and 75th percentiles, respectively. The bars represent values more than 1.5 times the interquartile range from the border of each box. The *p* values were calculated using the Wilcoxon rank-sum test. Two-sided *p* values were calculated. **c** GSEA was used to assess the upregulated and downregulated pathways of differentially expressed genes among before CAR-T cell therapy, during CAR-T cell expansion, and during disease progression targeting CD8 Teff

and CD8 Tex cells. **d** Trajectory analysis of CD8+ T cells. **e** CD8+ T cells were divided into three main states: in state 1, immunoactivated Teff was in equilibrium with exhausted Tex and Prolif; in state 2, Tex cells were the dominant cells; and in state 3, the proportion of Tex cells was significantly lower than that in state 2, but the proportion of Prolif cells with exhaustion characteristics was significantly increased. According to the distribution characteristics of each state at different time points, CD8+ T cells were mainly distributed in state 1 before CAR-T cell therapy, in state 2 during CAR-T cell expansion, and in state 3 during disease progression. **f** The expression of genes related to the effector molecular and transcription factors of CD8 Teff cells as well as the inhibitory molecules and transcription factors of CD8 Tex cells was assessed. Data are shown as mean ± s.e.m. Statistical analysis was performed using two-way ANOVA with Tukey's multiple comparison tests.

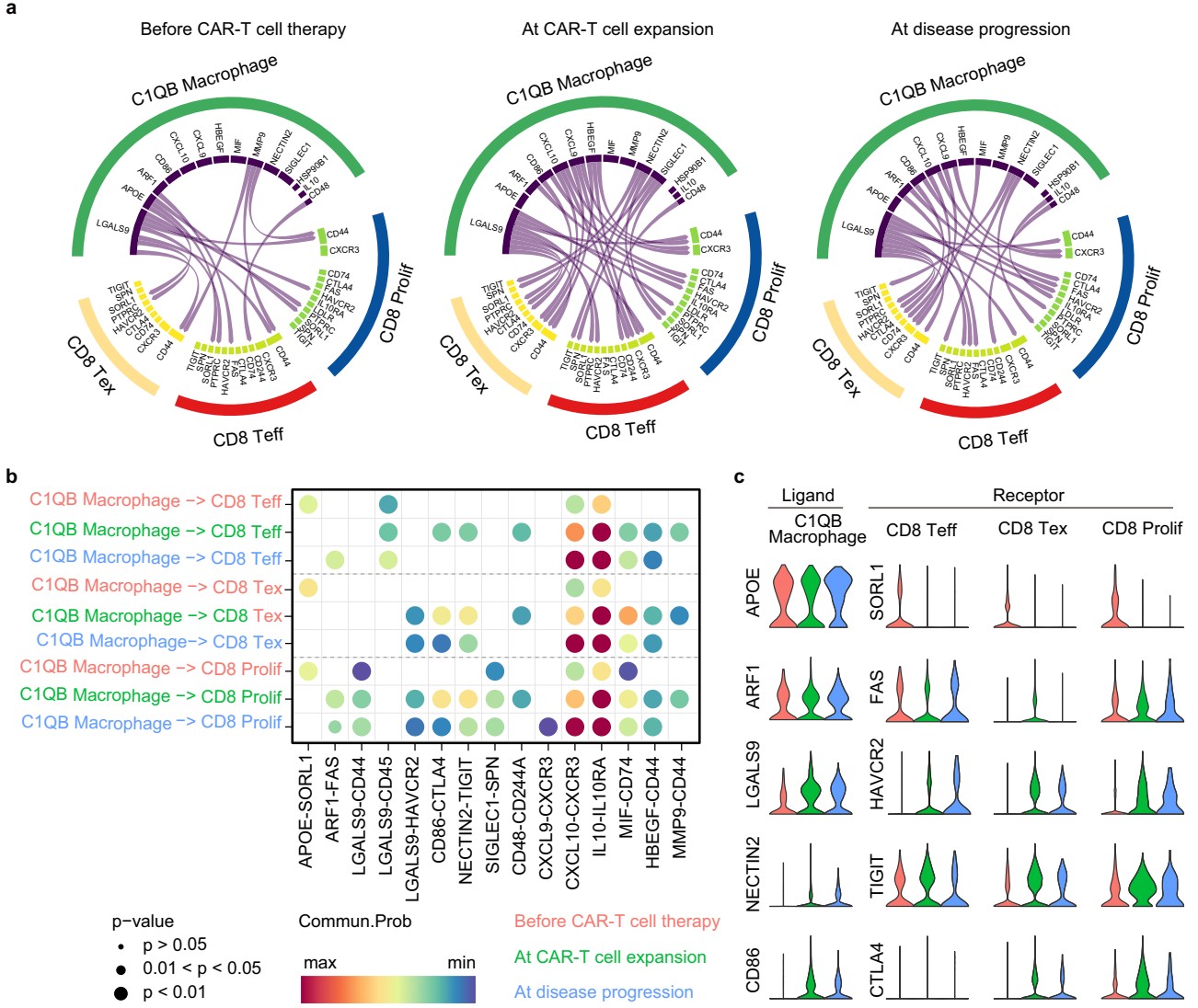

**Fig. 5 | T cell exhaustion triggered by ligand-receptor interactions between M2 macrophages and CD8+ T cells. a** The Cellchat software was used to assess differences in cell communication between CD8+ T cells and M2 macrophages before CAR-T cell therapy, during CAR-T cell expansion, and during disease progression in patients with PD. **b** The ligand-receptor interactions of M2 macrophages and CD8 T cells were predicted before CAR-T cell therapy, during CAR-T cell expansion, and during disease progression. *p* values are computed from one-sided permutation test. **c** Expression of ligands (APOE, ARF1, LGALS9, NECTIN2, and CD86) on M2 macrophages, and expression of receptors (SORL1, FAS, HAVCA2, TIGIT, and CTLA4) on CD8+ Teff, CD8+ Tex, and CD8+ Prolif cells. *n* = 3 for before CAR-T cell therapy group, *n* = 3 for at CAR-T cell expansion group, and *n* = 2 for at disease progression group in patients with PD.

M2-polarized macrophages on CD8+ T cell exhaustion in vitro. PBMCs were cocultured with medium, control cells (THP1 cells treated with PMA), M2 macrophages, and ABCA1-knockdown M2 macrophages for 48 h, followed by coculture with DB cells to mimic the TME. The

cholesterol levels in cells and medium as well as the percentage of T cells and levels of immune checkpoints were quantified (Fig. 6a). The M2-PBMC group exhibited a significantly lower cholesterol level than the control-PBMC group and medium group; however, a concomitant

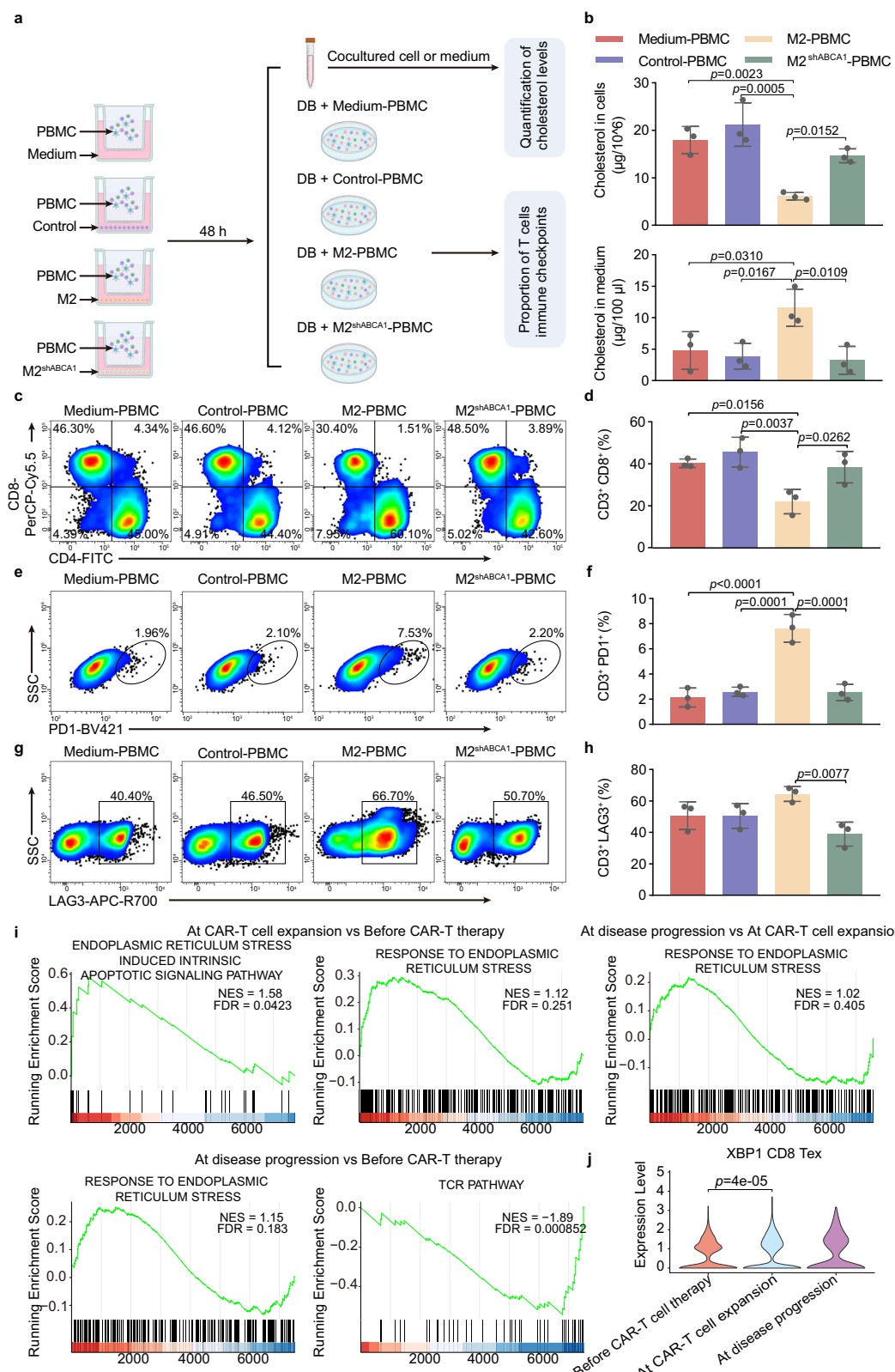

**Fig. 6 | T cell exhaustion induced by cholesterol efflux from M2 macrophages through TCR signaling inactivation. a** Flowchart of the in vitro assay. In brief, PBMCs were cocultured with medium, control cells, M2 cells, and ABCA1-knockdown (M2$^{shABCA1}$) M2 cells for 48 h, followed by coculture with DB cells to mimic the TME. Then, cholesterol levels in cells and medium as well as the percentages of T cells and immune checkpoint genes were quantified. **b** Quantification of total cholesterol levels in cells and medium. Flow cytometry analysis of the

percentages of CD8$^+$ T cells (**c, d**), PD1$^+$ cells (**e, f**), and LAG3$^+$ T cells (**g, h**). **i** GSEA of the upregulated and downregulated pathways of differentially expressed genes among before CAR-T cell therapy, during CAR-T cell expansion, and during disease progression targeting CD8 Teff cells. **j** Expression levels of the ER-stress-related gene XBP1 in CD8 Tex before CAR-T cell therapy, during CAR-T cell expansion, and during disease progression. Data are presented as mean ± s.e.m. Statistical analysis was performed using two-way ANOVA with Tukey's multiple comparison tests.

higher cholesterol level was detected in the M2-PBMC culture medium. ABCA1 inhibition (M2$^{shABCA1}$ PBMC group) restored normal cholesterol levels in both the M2-PBMC and coculture media (Fig. 6b). Furthermore, a significant reduction in the percentage of CD8$^+$ T cells (Fig. 6c, d) and an increase in the percentages of PD1$^+$ (Fig. 6e, f) and LAG3$^+$ T cells (Fig. 6g, h and Supplementary Fig. 8d) were detected in the M2-PBMC group, which were normalized by ABCA1 knockdown, suggesting that cholesterol induces the expression of T cell exhaustion-related genes. We co-cultured tumor cells and PBMC-derived M2 macrophages from DLBCL patients to mimic the TME. The cholesterol levels in cells and culture supernatant, the percentages of T cells, and levels of immune checkpoints were quantified, suggesting that macrophage cholesterol efflux induces T cell exhaustion. (Supplementary Fig. 7).

Cholesterol can induce T cell exhaustion by downregulating the TCR pathway and subsequently activating ER stress[24]. Consistent with this, the TCR signaling pathway was downregulated, whereas the ER-stress response pathway was upregulated in Teff and Tex cells during CAR-T cell expansion and disease progression (Fig. 6i). Subsequently, we evaluated ER-stress-related gene expression during CAR-T cell expansion and at disease progression. XBP1 expression was significantly upregulated during CAR-T cell expansion and disease progression (Fig. 6j). These results indicated that increased cholesterol efflux from C1QB macrophages induces immune checkpoint upregulation and T cell exhaustion through concomitant activation of XBP1/ER stress and inhibition of the TCR signaling pathway.

## Discussion

Despite the superiority of second-line CAR-T cell therapy over standard immunochemotherapy with or without ASCT[11,12], less favorable survival following CD19 CAR-T cell therapy in patients with primary refractory DLBCL compared with that in patients with relapsed DLBCL was reported in both TRANSFORM and ZUMA-7 trials. This multicenter phase I trial enrolling only patients with primary refractory DLBCL met the primary endpoint of manageable safety and demonstrated encouraging improvements in PFS and OS. Moreover, the median duration of CRS was longer in our study (10 days in this study vs. 7 days in other relma-cel studies) than in previous studies[19,25]. This is because we did not record the end of CRS until observing the disappearance of fever and other minor symptoms, such as fatigue, headache, and myalgia. Intensive monitoring, follow-up, and supportive treatments were conducted for patients with mild CRS symptoms, and tocilizumab was not administered for non-life-threatening grade 1 events. This was evidenced by the relatively lower rate of tocilizumab usage (25.0%) in our study. For your reference, we adhered to widely accepted approaches for CRS management in NHL, as outlined in the clinical trial protocol. These approaches have been adapted and modified from Lee 2014 and Neelapu SS 2017[31,32].

Consistent with a previous report, we found that the antitumor function of CAR-T is independent of IFN-γ production in DLBCL[33]. Notably, although SD/PD patients showed rapid expansion of CAR-T on day 1, they exhibited less durable response (90 days vs. 365 days). This may be attributed to the low level of long-T cells in SD/PD patients, which are crucial for generating long-lasting effect of CAR-T therapy[34,35]. Conversely, the early-time expansion of CAR-T cells was not associated with the therapeutic response of CAR-T[25]. Improvement of T cell effector and memory functions could be a potential strategy to increase the efficacy of CAR-T cells against tumors[36]. However, chronic exposure to antigenic stimulation subverts CD8$^+$ T cell differentiation into an exhausted state characterized by the loss of effector function and persistent expression of inhibitory receptors, subsequently impeding T cell survival and functions[37,38]. CD8$^+$ T cell exhaustion has been reported in various hematological malignancies, including acute myelocytic leukemia, acute lymphoblastic leukemia, chronic lymphocytic leukemia, multiple myeloma, and lymphomas[39]. In patients with

myeloma who received ASCT, the identified subset of exhausted and senescent CD8$^+$ T cells was associated with poor prognosis and predicted relapse[40,41]. CD8$^+$ T cell exhaustion triggered by the immunosuppressive TME represents a major obstacle in the efficacy of immunotherapy[42]. We studied their expression profile and interaction with TME using scRNA-seq, in order to reveal the mechanisms underlying the resistance to CAR-T cell therapy in patients with primary refractory DLBCL. We found that the T cell activation and TNF-α signaling pathways were downregulated in Teff cells of patients with PD before CAR-T cell therapy. Similarly, antigen receptor-mediated signaling pathways and T cell chemotaxis regulation were downregulated in the Tex subcluster of patients with PD, suggesting that the Teff and Tex subclusters were in a more immunosuppressive state in patients with PD before CAR-T cell therapy due to increased C1QB macrophages, triggering resistance to CAR-T cell therapy.

In line with our previous work[19,43], the percentages of multiple myeloid subpopulations were higher in SD/PD patients, including cDC1, cDC2, pDC, IL1B macrophage, and C1QB macrophage, with C1QB macrophage which is M2-polarized macrophages being the most predominant subset. Increased M2 macrophage infiltration was also reported in patients with refractory B-NHL who received anti-CD19 CAR-T cell therapy and was negatively associated with remission status as well as T cell function and proliferation inhibition[19,43,44]. Therefore, our results highlighted the vital roles of the pretreatment TME landscape in resistance to CAR-T cell therapy. Tumor progression is highly affected by C1QB macrophage-induced CD8$^+$ T cell exhaustion and T cell function and proliferation inhibition, highlighting the reprogramming of M2-polarized macrophages into a proinflammatory phenotype in the TME as a potential therapeutic approach to enhance the effects of CAR-T cells on tumor cells.

Cholesterol accumulation in the TME is essential for triggering T cell exhaustion[24,45]. In a murine melanoma model, adoptively transferred cholesterol-acquired CD8$^+$ T cells exhibited high levels of immune checkpoints genes and reduced antitumor activity[24]. In a mouse mesothelin-expressing pancreatic carcinoma model, the inhibition of cholesterol acyltransferase 1 (ACAT-1) in CAR-T cells resulted in significant tumor regression[46]. Cholesterol levels were reduced after CAR-T cell infusion in several hematological malignancies, suggesting that cholesterol metabolism may impact CAR-T cell therapy[47]. In ovarian cancer cells, membrane cholesterol efflux drives TAM-mediated tumor progression[26,48]. In the present study, the coculture of DLBCL cells with M2 macrophages triggered the efflux of cholesterol from M2 macrophages into the TME and reduced the cytomembrane cholesterol levels of M2 macrophages. Besides, C1QB macrophages of SD/PD patients exhibited pronounced enrichment of "cholesterol efflux" pathway, which was considered to facilitate TAM-mediated tumor progression[26,48]. In line with our observations, ovarian cancer cells have been reported to promote membrane cholesterol efflux and depletion of lipid rafts from macrophages[26,48]. Increased cholesterol efflux has been shown to promote IL-4-mediated reprogramming, including the inhibition of IFNγ-induced gene expression[26].

Furthermore, knockdown of ABC transporters was found to suppress cholesterol efflux, revert the tumor-promoting functions of M2 macrophages, and inhibit tumor progression, highlighting the impact of metabolic alterations in resistance to CAR-T cell therapy[26]. In melanoma, T cell exhaustion is driven by increased cholesterol uptake in CD8$^+$ tumor-infiltrating lymphocytes (TILs) through activation of ER-stress responses[24]. Inhibition of cholesterol esterification and acceleration of lipid catabolism significantly enhance the effector T cell functions and antitumor activity of CD8$^+$ TILs[49]. In addition, TME-derived cholesterol induces TIL dysfunction by activating XBP1[50]. Consistent with this, we found that the ER-stress pathway and XBP1 expression were significantly increased during CAR-T cell expansion and disease progression in patients with PD. This finding provides

insights into a potential antitumor therapeutic strategy targeting cholesterol efflux or upstream signaling pathways regulating cholesterol metabolism.

Based on ligand-receptor interactions, crosstalk of C1QB macrophages and CD8[+] T cells was examined. The interactions related to metabolism (AFR1-FAS), cytokines [CXCL9-CXCR3[51,52] and CXCL10-CXCR3[51], and immune checkpoints (LGALS9-HAVCR2, CD86-CTLA4[53], and NECTIN2-TIGIT) were enhanced during disease progression, indicating their involvement in T cell exhaustion during CAR-T cell therapy. It has been reported that ARF1-FAS interaction promotes lipid metabolism and lipid raft accumulation[54]. Moreover, macrophages can interact with CD8[+] T cells through the immune checkpoint-related pairs LGALS9-HAVCR2, CD86-CTLA4, and NECTIN2-TIGIT, which are known to induce T cell exhaustion in DLBCL[55]. In particular, Tim-3, encoded by *HAVCR2*, binds to Galectin-9, encoded by *LGALS9*, to mediate CD8[+] T cell response inhibition[56], immune escape of human leukemia cells[57], and TAM differentiation toward an immunosuppressive phenotype[58]. The CD86-CTLA4 interaction is also implicated in T cell response inhibition[59,60], and CTLA4 blockade is considered to be a promising approach in cancer immunotherapy[61]. NECTIN2-TIGIT plays critical roles in the inhibition of cytotoxicity, granule polarization, and cytokine secretion in NK cells[62–65]. Dual PD-1/TIGIT blockade increases tumor-specific CD8[+] T cell expansion and function and promotes tumor rejection in mouse tumor models[66,67]. The above lines of evidence can explain why the interactions of these pairs were enhanced during disease progression in patients with PD, owing to their negative effects on T cell function.

In summary, CAR-T cell therapy was safe and effective in patients with primary refractory DLBCL. Combining cholesterol-reducing agents or antibodies targeting HAVCR2 or TIGIT with CAR-T cell therapy could be a promising treatment option to improve the antitumor effect of CAR-T cell therapy. The complexity of TME may necessitate a combination of immunotherapy targeting different immune checkpoint pathways as mechanism-based DLBCL immunotherapy.

## Methods

### Ethics approval and consent to participate
The use of samples for this study was approved by ethical permission from the National Research Ethics Committee (REC: 2019-112). Appropriate approvals and informed written consent for study participation were obtained and the study was performed in accordance with the Declaration of Helsinki.

### Study design and patients
This phase I, single-arm, open-label, multicenter clinical trial was designed to preliminarily evaluate the safety and efficacy of relma-cel in adult patients with primary refractory DLBCL. Eligible patients met the following criteria: (1) age ≥18 years; (2) histologically confirmed DLBCL in accordance with the World Health Organization classification; and (3) achieved SD after at least three cycles of first-line treatment consisting of rituximab and anthracycline, or PR with residual lesions confirmed by biopsy after six cycles of first-line treatment, or PD following any cycle of first-line treatment. The patients were excluded if they received second-line or subsequent therapies or were diagnosed with CD19-negative lymphoma. A full list of patient inclusion and exclusion criteria is provided in the Clinical Trial Protocol in the Supplementary Information file. All patients provided written informed consent. The study was registered at www.chinadrugtrials.org.cn (CTR20200376).

### Relma-cel preparation and intravenous infusion
Autologous PBMCs were collected from each participant through leukapheresis. T cells were specifically enriched via apheresis using CD4 and CD8 microbeads and then activated using CD3/CD28 microbeads. The activated T cells were transduced ex vivo with replication-deficient self-inactivating (SIN) lentiviral vectors containing the CAR transgene and expanded in cell culture. Lymphodepletion preconditioning was accomplished with fludarabine at 25 mg/m$^2$/day and cyclophosphamide at 250 mg/m$^2$/day through intravenous injection for 3 days from day −7 to −2, followed by intravenous infusion of $100 \times 10^6$ relma-cel (JWCAR029; Shanghai Ming Ju Biotechnology Co., Ltd.) on day 0. Bridging chemotherapy was administered depending on patients' tumor burden as assessed by investigators.

### Outcomes
The primary endpoints were TEAEs and the type, frequency, and severity of laboratory examination abnormalities. AEs were assessed based on the National Cancer Institute Common Terminology Criteria for Adverse Events (NCI-CTCAE) version 5.0. The CRS was graded according to the 2014 criteria by Lee et al.[31]. NT was graded according to the NCI-CTCAE. Safety events were monitored from ICF obtained through 2 years of follow-up. The secondary endpoints included CRR and ORR at 1 and 3 months as well as PFS, OS, and pharmacokinetic (PK) parameters during disease progression. the detection and qualification of C-reactive protein, ferritin, T cell subsets, anti-CAR T antibodies, and serum cytokines (IFNα, IFNγ, IL-2, IL-5, IL-6, IL-8, IL-10, IL-2R, TNFα, and IL-1β), the concentrations of interleukin in plasma samples were measured by multiple microsphere flow immunofluorescence. Patients were followed up for 24 months to assess the safety and disease progression at 2, 3, 6, 9, 12, 18, and 24 months after CAR T infusion. C-reactive protein was detected using Beckman C-reactive protein reagent (catalog number: 447280) via AUC5800 Chemistry Analyzer, ferritin was detected using Beckman ferritin reagent (catalog number: 447160) via Unicel DXI 800 Access Immunoassay System, and IL-2R were detected using Simens IL-2R reagent (catalog number: LKIP1) via IMMULITE 1000. Detailed information about other cytokines were listed in Supplementary Table 4. More details on the secondary endpoints are found in Supplementary Table 3.

### In vivo cytotoxicity assays
The cytotoxicity of CAR-T cell therapy was assessed using a standard calcein release assay. Target cells (Nalm-6) were stained with Calcein-AM (Life Technologies, Invitrogen) for 45–90 min at 37 °C in a 5% $CO_2$ incubator. Labeled Nalm-6 (30,000 cells in 100 µl) and effector cells (100 µl) were coincubated at an effector-to-target (E:T) ratio of 20:1 for 4 h at 37 °C. Maximum and spontaneous release controls were set up in three replicates using 1% Triton X-100 and plain medium, respectively. Fluorescence intensity was measured using a BioTek Synergy 2 plate reader (excitation: 485 nm, emission: 530 nm). Percent specific lysis was calculated using the following formula: [(Test release − Spontaneous release) / (Maximum release − Spontaneous release)] × 100.

### In vivo CAR-T cell detection
The persistence of CAR-T cells was measured using the blood samples of patients who received CAR-T cell therapy before and at predetermined intervals after CAR-T cell infusion until the time of first below the limit of quantification (BLQ, 125 copies/µg). Quantitative RT-PCR was used to detect integrated transgene sequences using the following primers: forward: 5′-CCG TTG TCA GGC AAC GTG-3′ and reverse: 5′-AGC TGA CAG GTG GCA AT-3′.

### Single-cell suspension preparation
Tumor samples were digested using the GEXSCOPE Tissue Preservation Solution (Singleron Biotechnologies, USA) at 37 °C for 15 min, followed by separation of cells from cell debris and other impurities using a 40-µm sterile strainer (Corning, USA). To remove red blood cells, 2 ml of GEXSCOPE Red Blood Cell Lysis Buffer (Singleron Biotechnologies, USA) was mixed with the cell suspension and incubated at 25 °C for 10 min. The mixture was centrifuged at $500 \times g$ for 5 min and the cell pellet was resuspended in PBS. The cells were counted with a TC20 automated cell counter (Bio-Rad, USA).

## Single-cell RNA library preparation and sequencing

Single-cell RNA libraries were prepared using the 10× Chromium Single Cell platform using a Chromium Single Cell 3′ Library, Gel Bead and Multiplex Kit, and Chip Kit (10× Genomics, Pleasanton, CA, USA). The loaded FACS-sorted viable hCD45[+] cell numbers ranged from 8000 to 16,500, with final viability of >80%, aiming for 2000–10,000 single cells per channel. Following the generation of single-cell gel bead-in-emulsions (GEMs), reverse transcription and amplification were performed. Then, amplified cDNAs were purified and sheared. Purified libraries were sequenced on the NovaSeq 6000 platform (Illumina, San Diego, CA, USA) or the BGI MGISEQ-2000 platform (Shenzhen, China) as per the manufacturer's protocol.

## Single-cell RNA-seq data preprocessing and quality control (QC)

The Cell Ranger software pipeline (version 6.0.1) provided by 10× Genomics[68] was applied to demultiplex cellular barcodes, map reads to the GRCh38 reference assembly using the STAR aligner, and produce a feature-barcode unique molecular identifier (UMI) matrix. Ten single-cell RNA-seq libraries were sequenced to average 381,854,593 (324,594,163–479,610,062) paired-end reads per cell with 52.4% (29.5–62.0%) sequencing saturation. We processed the UMI count matrix using the R package Seurat (version 4.0.4)[27]. As a QC step, we first filtered out genes detected in more than three cells and cells where fewer than 200 genes had nonzero counts. To remove likely doublet captures, we further excluded cells with total UMI counts >60,000 and the number of detected genes was <200 or > 6000. Following removing cells with high expression of mitochondrial genes, we further discarded low-quality cells where >20% of the counts belonged to mitochondrial genes. We also applied the DoubletFinder (version 2.0.3) R package for each library separately to identify potential doublets[69]. The expected doublet rate was set to be 10.0%, and cells predicted to be doublets were filtered. After quality control, a total of 84,881 single cells from 10 libraries were remained for downstream analysis.

## Single-cell RNA-seq normalization, batch effect correction, dimensionality reduction, and unsupervised clustering

After QC and filtration, the feature-barcode matrices of each library were processed using the Seurat (version 4.0.4) R package for normalization, highly variable feature identification, scaling, and linear dimensional reduction[27]. First, all 21 libraries were combined using the Seurat merge function. Library size normalization was performed in Seurat on the filtered matrix to obtain the normalized count by NormalizeData. Features that exhibited high cell-to-cell variation in the dataset were identified via FindVariableFeatures, and a total of 3000 highly variable features were returned. Features were then centered and scaled using ScaleData, and principal component analysis (PCA) was conducted using RunPCA. Next, to integrate cells into a shared space from different datasets for unsupervised clustering, we used the harmony algorithm to perform batch effect correction[70]. A PCA matrix with 30 components using such informative genes was fed into the RunHarmony function implemented in the R package harmony (version 1.0). For visualization, the dimensionality of each dataset was further reduced using the UMAP by the Seurat function RunUMAP. For unsupervised clustering, a K-nearest neighbor based on the Euclidean distance in PCA space was first calculated and a shared nearest neighbor graph was constructed using FindNeighbors. Then, modularity optimization techniques using the Louvain algorithm were applied via FindClusters to identify clusters of cells.

## Single-cell RNA-seq cell subset identification and annotation

The first round of clustering (resolution = 0.3) identified four major cell types, namely, B cells, T/NK cells, fibroblasts, and myeloid cells. In total, we identified four major cell types, namely, B cells (CD79A, CD79B, MS4A1, and CD19), CD4 and CD8 T cells (CD3D, CD3E, CD3G,

CD40, CD40LG, CD8A, and CD8B), NK cells (GNLY, NKG7, TYROBP, and PRF1), and fibroblast (ACTA2) and myeloid cells (CST3 and LYZ). To identify clusters within each major cell type, we performed a second round of clustering on T/NK and myeloid cells separately. The procedure for this second round of clustering was the same as that for the first round, starting from low-rank harmony output (30 components) on the highly variable genes chosen as described above, with resolution ranging from 0.1 to 0.9. We used the clustree (version 0.4.4)[19] R package to visualize and evaluate the above clustering results. Meanwhile, single cells expressing two sets of well-studied canonical markers of major cell types were labeled as doublets and excluded from the following analysis. T cells were divided into NK, CD8 effector T, CD8 exhausted T, CD8 proliferation T, CD4 Naïve T, Treg, CD4 memory T, CD4 Th1 like T cell. Myeloid cells were divided into 7 clusters. The cDC1 subcluster were characterized by marker gene CLEC10A and CD1C; the cDC2 subcluster were characterized by highly expressed CLEC9A and XCR1; the cDC3 subcluster were characterized by highly expressed LAMP3 and IDO1; the pDC subcluster expressed CLEC4C, TCF4 and IRF7. C1QB macrophages were characterized by C1QB, APOE and CD163 expression, whereas IL1B macrophages were characterized by IL-1B and FCN1 expression[50].

## Single-cell RNA-seq signature score

For gene scoring analysis, we compared different gene signatures in subpopulations using the Seurat AddModuleScore function[27]. The immunosuppressive signature score was defined as the average expression of a series of immune checkpoint inhibitors[28] and immunosuppressive molecules[29], including CD244, CD160, CTLA4, PDCD1, TIGIT, LAYN, LAG3, HAVCR2, CD274, CD47, CD96, ENTPD1, VSIR, BTLA, EBI3, IL2RB, IL2RA, and IL2RG, whereas the dysfunctional signature score[30] was defined by the average expression of LAG3, HAVCR2, PDCD1, PTMS, FAM3C, IFNG, AKAP5, CD7, PHLDA1, ENTPD1, SNAP47, TNS3, CXCL13, RDH10, DGKH, KIR2DL4, LYST, MIR155HG, RAB27A, CSF1, TNFRSF9, CTLA4, CD27, CCL3, ITGAE, PAG1, TNFRSF1B, GALNT1, GBP2, MYO7A, and TIGIT. We also included a list of transcription factors[71] that were related to T cell exhaustion, including BATF, BCL6, BHLHE40, BTLA, CD200, EOMES, ETV1, FOXP3, HIF4A, HOPX, ID2, ID3, IFI16, IKZF, IKZF3, NR4A1, NR4A2, NR4A3, PRDM1, RBPJ, SOX4, STAT3, TBX21, TCF7, TOX, TOX2, VDR, ZBED2, ZNF683, IFI16, DRAP1, and ETS1.

## Differential expression and Gene Ontology enrichment analysis

For cell cluster-specific marker genes, we performed two-sided unpaired Wilcoxon tests on all expressed genes (expressed in at least 20% of cells in either cluster of cells) using the Seurat FindAllMarkers function. We selected cell type-specific signature marker genes and visualized them with a dot heatmap plot or stacked violin plot. We also used the FeaturePlot to generate the gene expression feature plots, in which each cell was colored based on the expression level of the selected gene. The top 50 highly expressed genes of each cluster are shown in the heatmap plot. We chose cell cluster-specific DEGs with an absolute average fold change of >1.5 and adjusted p values of <0.05 and excluded mitochondrial genes or ribosomal genes from downstream enrichment analysis. In addition, we used the FindMarkers function to calculate differential expressed genes. We used a two-sided unpaired Wilcoxon test for genes expressed by at least 10% of cells in either group. Based on the differential expression from the single-cell gene expression data, significantly enriched Gene Ontology (GO) terms were acquired for each cluster using the R package clusterProfiler (version 4.1.4)[72].

## Developmental trajectory inference

To determine potential lineage differentiation between different T cell populations, we performed trajectory analysis using the Monocle 2 (version 2.18.0)[73] algorithm. A CellDataSet object was created using the newCellDataSet function with expressionFamily

set to be negbinomial.size. Dimensionality reduction was performed with the DDRTree algorithm and max_components parameters = 4, using the expression of the top 3000 highly variable genes detected as described above. The cell trajectory was then captured using the orderCells function, and the inferred cell trajectories were visualized using the plot_cell_trajectory function. To visualize genes whose expression levels changed along with the pseudotime trajectory, we used the plot_pseudotime_heatmap function and genes with an adjusted $p$ value of <0.01, fold change of >1.5, and belonging to the transcription factors. To detect genes that play an important role in cell fate decisions, we implemented branched expression analysis modeling to identify genes with branch-dependent expression. We used the plot_genes_branched_heatmap function to visualize genes with branch-dependent expression, where the genes were selected as either $q$ val < 0.05 and belonging to transcription factors or $q$ val < 1e−6 and belonging to the top 3000 highly variable genes. These genes were also used for GO enrichment analysis.

### Cell-cell ligand-receptor communication analysis

We used the Cellchat (version 1.1.3)[74] R package to identify and visualize the intercellular communication networks between T cells and C1QB macrophages from scRNA-seq data. The package contained 1939 pairs of well-curated ligand and receptor pairs, including 1199 pairs of secreted signaling interactions, 319 pairs of cell-cell contact interactions, and 421 pairs of extracellular matrix receptor interactions. We calculated interactions and interaction strength among different DLBCL groups across time and compared the overall information flow of each signaling pathway. The minimum number of cells required for cell-cell communication analysis in each cell group was set to be 10. We extracted significant cellular communication pairs with $p$ values < 0.05 and extracted the T cell exhaustion-related pathways. We then identified all the significant interactions (ligand-receptor pairs) for a list of T cell exhaustion-related pathways and visualized their average expression among different cell subtypes by ComplexHeatmap (version 2.6.2)[75] R package. For a given pair of ligand and receptor, we used a violin plot to visualize their expression levels across groups[76].

### Visualization and statistical analysis

The GO enrichment results were visualized as dot or bar plots using the ggplot2 (version 3.3.5) R package. Bar plot and pie plot were also generated using the ggplot2 (version 3.3.5) R package. Heatmaps were generated using the ComplexHeatmap (version 2.6.2) R package[77]. Unpaired two-sided Wilcoxon rank-sum tests were used for pair-wise comparisons. Statistical significance was accepted at $p < 0.05$. For all differential expression and gene set testing analyses, $p$ values were corrected for multiple testing using the Benjamini–Hochberg protocol. All statistical analyses were performed in R (version 4.0.5).

### Cell culture, ABCA1 knockdown, and CAR19 generation

Monocyte cell line (THP-1), DLBCL cell line (DB), acute promyelocytic leukemia cell line (HL60), and 293 T cells were obtained from American Type Culture Collection (ATCC, MA, USA). THP-1 and DB cells were maintained in RPMI-1640 medium (Thermo Fisher) supplemented with 10% fetal bovine serum (FBS) (Thermo Fisher). HL60 cells were grown in Iscove's Modified Dulbecco's Medium (DMEM) supplemented with 20% FBS. 293 T cells were maintained in DMEM supplemented with 10% FBS.

Human peripheral blood mononuclear cells (PBMCs) were obtained from healthy donors. PBMCs were isolated from blood buffy coats utilizing a density gradient. Subsequently, PBMCs were suspended in serum-free RPMI 1640 medium for 1 h to facilitate monocyte adhesion. The cultured PBMCs were maintained at a density of $2 \times 10^6$ cells/ml in 6-well plates for 24 h. To initiate macrophage differentiation, a final concentration of 50 ng/ml macrophage colony-stimulating

factor (M-CSF) was added to the culture medium. PBMC-derived macrophages (control cells) were obtained through induction after culturing for 6 days. To stimulate PBMC-derived M2 macrophages, IL-4 (20 ng/ml) was then added and incubated for 24 h[78].

To generate M1- and M2-polarized macrophages, $1 \times 10^6$ THP-1 cells were seeded into the upper wells of Transwell chambers (0.4-μm pores; Corning Inc., Corning, NY, USA) and treated with 320 nM PMA for 24 h to generate M1 macrophages or treated with 320 nM PMA for 6 h, followed by supplementation of PMA (320 nM), IL-4 (20 ng/ml), and IL-13 (20 ng/ml) cocktail for 18 h, to generate M2 macrophages.

For ABCA1 knockdown, M2 macrophages were transfected with shRNA targeting ABCA1 (shRNA-ABCA1) (Gene Co., Ltd, Shanghai, China) using Lipofectamine 3000 (Invitrogen, Carlsbad, CA, USA) in accordance with the manufacturer's instructions.

For pharmacological approach to promote cholesterol efflux in M2 macrophage, we used 9-cis-Retinoic acid (9cRA) (MedChemExpress LLC, Shanghai, China) in accordance with the manufacturer's instructions. For CAR T generation, a dual-targeted CAR lentiviral construct incorporating CD19 from clone FMC63 with the 4-1BB costimulatory and CD3z signaling domains was used. The lentivirus was propagated in 293 T cells, cryopreserved at −80 °C, and thawed immediately before transduction. CD3$^+$ T cells were activated by anti-CD3/CD28 Dynabeads (Gibco, catalog no: 40203D). Cells were cultured in X-VIVO 15 medium (Lonza, catalog no.: 04-418Q) supplemented with 100 U/ml IL-2 and transduced with lentiviruses 1 day after stimulation at a multiplicity of infection of 2. The Dynabeads were removed on day 4.

### Transwell assay

Various types of cells were cocultured using Transwell assay. In detail, 1000 PBMC/CAR 19 cells were seeded on the top of the Transwell membrane (1-μm pore size, 662610; Greiner Bio-One, Alphen aan den Rijn, South Holland), whereas 500 control cells, M2 cells, or ABCA1-knockdown M2 cells (M2$^{shABCA1}$) were cultured in the lower compartment in 24-well plates for 48 h.

### Total cholesterol extraction and quantification

For cholesterol extraction, cells ($1 \times 10^6$) or medium (100 μl) were mixed vigorously with methanol/chloroform (1:2, v/v) at room temperature for 2 h. The organic phase containing cholesterol was collected, and the solvent was allowed to evaporate under vacuum. Finally, cholesterol in each sample was quantified using the Amplex Red cholesterol assay kit (Invitrogen, Carlsbad, CA, USA) according to the manufacturer's instructions.

### In vitro cytotoxicity assay

DB-luci tumor cells were generated and used in a luciferase-based CTL assay[79]. In brief, DB tumor cells were transduced with a firefly luciferase-encoding lentivirus to generate parental cells for target cell line preparation. Target cells were resuspended at a concentration of $1 \times 10^5$ cells/mL in X-VIVO 15 medium and incubated with different ratios of CAR-T cells (e.g., 16:1 and 8:1) overnight at 37 °C. Samples (100 μl) were transferred to a 96-well white luminometer plate and mixed with 100 μl of the substrate solution. Then, luminescence was immediately determined. The results are reported as the percentage of killing based on the luciferase activity in the wells with tumor cells but no CAR-T cells [% killing ¼ 100 − (relative light units (RLU) from well with effector and target cell coculture) / (RLU from well with target cells) × 10)].

### Flow cytometry

Cells were prewashed with PBS and incubated with antibodies for 30 min on ice. After washing twice, resuspended in 500 μl cellular preservation fluids were run on BD LSRFortessa (BD Biosciences, Franklin Lake, NJ, USA) and analyzed using FlowJo software. The following antibodies were used: FITC anti-human CD4, BV650 anti-human

CD4, BV421 anti-human PD1, APC-R700 anti-human LAG3, BV786 anti-human CD3, PerCP-Cy5.5 anti-human CD8, APC anti-human CD206, PE-Cy7 anti-human CD68, BV605 anti-human CD11b, PE anti-human CD45RO, APC anti-human CD62L, PE anti-human LAG3, PE anti-human TIM3, PE anti-human CD107a, BV786 anti-human CD8 (all from BD Biosciences). Detailed information about antibodies are listed in Supplementary Table 5.

## Immunofluorescence

The Vybrant Alexa Fluor 488 lipid raft labeling kit (Thermo Fisher Scientific, Waltham, MA, USA) was used to assess the cytomembrane cholesterol levels in M2 macrophages according to the manufacturer's instructions. THP-1 cells or M2 macrophages were grown and stimulated in Lab-Tek chambered slides. After washing with serum-free RPMI-1640 medium, the cells were incubated with Alexa488-conjugated cholera toxin subunit B (CTB) at 4 °C for 10 min, followed by cross-linking with an anti-CTB antibody at 4 °C for 15 min. Subsequently, the cells were fixed with 4% Antigenfix (Diapath S.P.A., Martinengo BG, Italy) for 10 min on ice, and nuclei were stained with DAPI. Fluorescence intensity was measured via confocal microscopy (LSM780, Carl Zeiss GmbH, Oberkochen, Germany; or SP5X, Leica Microsystems, Wetzlar, Germany) and analyzed in a blinded manner with FIJI software.

## Statistical analysis

This phase I open-label single-arm study does not involve randomization due to its design. Given the small sample size in this study, the results from statistical tests are not controlled for type I error or adequately powered to draw inferential conclusions.

Descriptive statistics are used in displaying the results of the primary endpoint and secondary endpoints. Kaplan–Meier curves with log-rank test were used to assess the PFS and OS. D'Agostino-Pearson omnibus normality test, $t$-test, or one-way ANOVA was used to analyze normally distributed data, whereas nonparametric test (Wilcoxon signed-rank test or Friedman test) was used for non-normally distributed data. GraphPad Prism v7 (USA) and R (version 4.0.5) were used for statistical testing and visualization in this study. One/two-sided tests of hypotheses were conducted at $\alpha$ of 0.05. The detailed descriptions of the statistical tests used were provided in the legends of the corresponding figures. Figures were created using BioRender and Figdraw.

## Reporting summary

Further information on research design is available in the Nature Portfolio Reporting Summary linked to this article.

## Data availability

The raw sequencing data generated during this study have been deposited in the Genome Sequence Archive in National Genomics Data Center, China National Center for Bioinformation/Beijing Institute of Genomics, Chinese Academy of Science under accession number "GSA-Human: HRA006798". These data are under controlled access by human privacy regulations and are only available for research purposes. Access to the data can be granted following approval from the Data Access Committee of the GSA-human database, as detailed at https://ngdc.cncb.ac.cn/gsa-human/document/GSA-Human_Request_Guide_for_Users_us.pdf. Data are accessible to researchers who meet the criteria for access as defined by the GSA-human database guidelines. Access requests are usually processed within approximately 4 weeks and data will be available for 3 months once access is granted. The remaining data are available within the Article, Supplementary Information, or Source Data file. Source data are provided with this paper.

## Code availability

The source code for data cleaning and analysis is accessible for scientific research purposes on GitHub (https://github.com/MikaQiao/scDLBCL) and Zenodo (https://doi.org/10.5281/zenodo.10720059).

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

## Acknowledgements

This study was funded in part by research funding from the National Natural Science Foundation of China (81830007, 82130004, 81600155, 81670716, 82300169 and 82300210), Clinical Research Plan of SHDC (2020CR1032B), Chang Jiang Scholars Program, Shanghai Rising-Star Program (19QA145600), Municipal Human Resources Development Program for Outstanding Young Talents in Medical and Health Sciences in Shanghai (2017YQ075), Talent (Class A) of Guangci Excellence Youth Plan (GCQN-2019-A16), Clinical Research Plan of Shanghai Hospital Development Center (SHDC2020CR1032B), Shanghai Municipal Education Commission Gaofeng Clinical Medicine Grant Support (20152206 and 20152208), Shanghai Sailing Program (22YF1426400 and 22YF1425500), Clinical research project by Shanghai Jiao Tong University School of Medicine (CARTFR-05, KY2023727, YW20220022), Samuel Waxman Cancer Research Foundation and Innovation Technology Launch Plan of Guangci, Shanghai Clinical Research Center for Cell Therapy (23J41900100). This clinical trial was sponsored by JW Therapeutics (Shanghai) Co., Ltd., who participated in the design of this study and the manufacture of CAR-T cells. We are grateful to the support staff there.

## Author contributions

Z.X.Y., W.L.Z., and Z.S.Z. contributed to the conception and design of the study. Y.D. conducted the experiments, analysis and interpretation of the data and drafted the manuscript. N.Q. and Y.D. conducted analysis and interpretation of the data. Y.L.Z., W.W., Y.Z., L.W., S.C. and P.P.X. contributed to reagents/materials/analysis tools. Z.X.Y., Y.D., N.Q., L.S.S. and W.L.Z. contributed to writing and final drafting of the manuscript. All authors have read and approved the final version of this manuscript.

## Competing interests

The authors declare no competing interests.

## Additional information

Zi-Xun Yan[1,4], Yan Dong[1,4], Niu Qiao[1,4], Yi-Lun Zhang[1], Wen Wu[1], Yue Zhu[1], Li Wang[1], Shu Cheng[1], Peng-Peng Xu[1], Zi-Song Zhou[2], Ling-Shuang Sheng ®[1] ✉ & Wei-Li Zhao ®[1,3] ✉

[1]Shanghai Institute of Hematology, State Key Laboratory of Medical Genomics, National Research Center for Translational Medicine at Shanghai, Ruijin Hospital Affiliated to Shanghai Jiao Tong University School of Medicine, Shanghai 200025, China. [2]JW Therapeutics (Shanghai) Co. Ltd, Shanghai 200025, China. [3]Pôle de Recherches Sino-Français en Science du Vivant et Génomique, Laboratory of Molecular Pathology, Shanghai 200025, China. [4]These authors contributed equally: Zi-Xun Yan, Yan Dong, Niu Qiao. ✉e-mail: sls12280@rjh.com.cn; zhao.weili@yahoo.com

