## [Peer Review File · Nature Communications]

Cholesterol efflux of M2 macrophage in CAR-T cell therapy resistance: A phase I study of primary refractory DLBCL (JWCAR029-003)RESPONSE TO REVIEWERS' COMMENTS

Response to Reviewer 3

Reviewer #3:

The authors have addressed the comments of this reviewer satisfactorily mostly, except that the following information needs to be provided.

1). Please spell out 9cRA in the Results and Figure legend and point out in the Results that 9cRA has been previously used to promote cholesterol efflux in

Response: First, I would like to extend my heartfelt appreciation to the reviewer for the extensive review and insightful recommendations. Your professional insights are immensely valuable in helping us improve our article. The details of 9cRA in the results and figure legend has been added. Please refer to lines 222-223 and 1109 in the revised manuscript.

2). Please include the information on the use of 9cRA in the Methods.

Response: Thank you for your suggestion. The methodology pertaining to the use of 9cRA has been detailed in lines 682-684 in the revised manuscript.

3). Statistics information is missing for the lower right panel showing the % of lysis of Supplementary Figure 4.

Response: We thank the reviewer for your helpful comment. Statistical information has been added in lines 1117-1119 in the revised manuscript.

4). For Supplementary Figure 6, please provide information, in the Methods, on the "Control" in the four different PBMC samples. Also, please provide information on the source(s) the PBMCs isolated, such as the individual people (normal or with disease, age, sex,

Response: Thank you for your suggestion. The term "Control" in the coculture system refers to the coculturing of PBMCs with control cells, which are macrophages derived from PBMCs and induced only by M-CSF, as shown in Supplementary Figure 6 at lines 670 – 672. The PBMCs were obtained from healthy donors aged between 20 to 30 years, with no gender restriction. Please refer to lines 664-665 in the revised manuscript.

5). Please provide WB data showing the ABCA1 KD efficiency and the statistics method(s).

Response: Thank you for your suggestion. The expression of ABCA1 in M2 and M2 ABCA1-knockdown macrophages by western blot were evaluated using the Image J software (Figure 1 below). statistical comparisons were made using a paired sample t-test.

Figure 1 The expression of ABCA1 in M2 and M2 ABCA1-knockdown macrophages. Fold changes are shown below the gel normalized to Actin, which was assigned a value of 1.00.

RESPONSE TO REVIEWERS' COMMENTS

Response to Reviewer 1

Reviewer 1 expert in biostatistics

Paper Review:

Summary:

The paper presents a Phase I trial assessing the efficacy and safety of CAR-T cell therapy in primary refractory DLBCL patients. The authors assert that CAR-T cell therapy shows promise in enhancing the anti-tumor effect in DLBCL patients.

Major Comments:

A notable concern in this study is the absence of support for these claims through power analyses or sample size calculations within the paper. The final sample size employed for drawing these conclusions comprises only 11-12 patients, which appears smaller than the customary sample size of approximately 20. It would greatly benefit the study to provide insights into how the power analysis was conducted during the study's design phase.

Response: First, I would like to extend my heartfelt appreciation to the reviewer for the extensive review and insightful recommendations. Your professional insights are immensely valuable in helping us improve our article.

This clinical study reflects an exploratory phase I clinical trial with safety as the primary endpoint, and detecting significant safety issues associated with treatment is the primary focus. Notably, during this exploratory study, we found interesting results using single-cell RNA sequencing in helping us understand the potential mechanism of CAR-T resistance in primary refractory DLBCL. This is why the manuscript was based on a relatively smaller sample size. The efficacy of Relma-cel CAR-T therapy in first-line, primary refractory DLBCL remains uncertain. For ethical reasons, ethics committees typically impose sample size limitations in phase I clinical trials to protect patients' rights and well-being. To conduct preliminary observation and efficacy assessment in early trials, this sample size restriction is employed, minimizing potential patient risks.

Additionally, in comparison to traditional safety-focused 3 + 3 clinical trials, which typically allocate a maximum of six patients to each dose level, our sample size is 12, which is reasonable to ensure detection of significant safety issues associated with treatment. To validate the

conclusions drawn from the exploratory phase I trial, a related confirmatory study with a larger sample size is ongoing (NCT06093841).

Minor Comments:

There are a few additional minor points to address:

- Lines 706-708: The statement made here is somewhat perplexing. It appears that all statistical tests were carried out under the assumption of a normal distribution. Is this perhaps a typographical error?

Response: We sincerely appreciate the reviewer's comments and sincerely apologize for the typographical error in the submission. In our resubmitted revised manuscript, this typographical error has been corrected from "normal distribution" to "non-normal distribution." Please refer to lines 760 for the revision.

- Another minor issue pertains to the selection of certain algorithm parameters, such as those for DDRTree and clustering. It would be helpful to clarify how these algorithm parameters were determined and fixed. This would help ensure the reproducibility of the work.

Response: We thank the reviewer for highlighting the issue on the selection of certain algorithm parameters. Discriminative Dimensionality Reduction with Trees (DDRTree) is a method employed in the analysis of single-cell RNA sequencing (scRNA-seq) data. DDRTree is categorized as a manifold learning algorithm, specifically designed for identifying and analyzing developmental trajectories or pseudotime ordering within scRNA-seq data. Reducing the high-dimensional gene expression data to a lower-dimensional space while preserving the underlying biological structure is its primary objective. The DDRTree is widely recognized for its superior power, accuracy, and robustness in single-cell trajectory analysis, and has become the default method in the monocle R package.

In the context of DDRTree, "max_components" typically denotes the maximum number of dimensions retained after the dimensionality reduction process. By specifying "max_components = 4," we are instructing DDRTree to reduce the data to a space with a maximum of four dimensions.

These parameter settings were selected according to the following relevant published literature 1-3.

References

[1] Li, N. et al. Memory CD4(+) T cells are generated in the human fetal intestine. *Nat Immunol* 20, 301-312, doi:10.1038/s41590-018-0294-9 (2019).

[2] Luo, Y. et al. Single-cell transcriptomic analysis reveals disparate effector differentiation pathways in human T(reg) compartment. *Nat Commun* 12, 3913, doi:10.1038/s41467-021-24213-6 (2021).

[3] Saelens, W., Cannoodt, R., Todorov, H. & Saeys, Y. A comparison of singlecell trajectory inference methods. *Nat Biotechnol* 37, 547-554, doi:10.1038/s41587-019-0071-9 (2019).

- Is the code used for the analyses open-source? A link would be helpful.

Response: We appreciate the reviewer's interest in the code utilized for our analyses. In our study, we are committed to promote transparency and reproducibility. To facilitate access to the code associated with the current submission, we have made it available on GitHub at the following link: <https://github.com/MikaQiao/scDLBCL>. Additionally, we have included this link in our revised manuscript, stating, "The source code for data cleaning and analysis is accessible for scientific research purposes on GitHub via <https://github.com/MikaQiao/scDLBCL>". Please refer to lines 965-968 for the revision.

Reviewer 2 expert in haematological malignancies

Thank you for the opportunity to review the manuscript by Yan et al entitled "Cholesterol efflux of M2 macrophage in CAR-1 T cell therapy resistance: A phase I study of primary refractory DLBCL (JWCAR029-003)". I enjoyed reading the story and believe it adds new and interesting information to the conversation about tumor associated features mediating resistance to CAR-T for DLBCL. Some grammatical features require attention, however, the message is clear: M2-mediated cholesterol efflux plays a role in T cell function within the DLBCL environment and is associated with response in this trial. I have the following comments which I believe should be addressed:

- 1. The clinical trial itself is interesting in that only Primary Refractory patients were enrolled. Was this data reported in another peer-reviewed manuscript? If yes, it should be cited. If no, I believe the significance of the trial being for only those with Primary Refractory disease should be highlighted in the introduction as this nuance otherwise is not highlighted until the discussion where the authors correctly point out that patients with refractory disease fare with CAR-T than those with relapsed disease. Unless this nuance is clear the reader could incorrectly assume this CAR-T is inferior (based upon CR/ORR rates on line 97) as compared to liso-cel or axi-cel, where instead a more aggressive population was treated here.**

Response: We greatly appreciate your kind and helpful comments. As you have accurately emphasized, conducting direct comparisons with liso-cel or axi-cel would be inappropriate owing to the distinct patient populations included in our study. Clarifying this significant aspect early in our manuscript is indeed crucial to prevent any potential misinterpretations.

In response to your valuable suggestion, in the introduction section, we will ensure a stringent emphasis on this distinction, specifically within lines 57-62. We intend to emphasize that while CAR-T cell therapy has shown superiority in second-line treatments, recent trials such as TRANSFORM ¹ and ZUMA-7 ² have reported less favorable survival outcomes following CD19 CAR-T cell therapy in primary refractory patients, as compared to relapsed/refractory DLBCL patients. We deem that providing this context is essential, considering the high unmet medical need within the primary refractory population.

This manuscript has not been previously published or presented elsewhere and it is not currently under consideration by another journal.

References

[1] Kamdar, M. et al. Lisocabtagene maraleucel versus standard of care with salvage chemotherapy followed by autologous stem cell transplantation as second-line treatment in patients with relapsed or refractory large B-cell lymphoma (TRANSFORM): results from an interim analysis of an open label, randomised, phase 3 trial. *Lancet* 399, 2294-2308, doi:10.1016/S01406736(22)00662-6 (2022).

[2] Locke, F. L. et al. Axicabtagene Ciloleucel as Second-Line Therapy for Large B-Cell Lymphoma. *N Engl J Med* 386, 640-654, doi:10.1056/NEJMoa2116133 (2022).

2. The finding presented in Supplemental Figure 1d is interesting and agrees with ZUMA-1 where more IFN-g release upon in vitro stim of the product was associated with worse efficacy (Locke et al., *Blood Advances*, 2020, DOI:10.1182/bloodadvances.2020002394) and data demonstrating that IFN-g from CAR-T cells is dispensable for effectiveness in hematologic malignancies (Larson et al, *Nature*, 2022, DOI:10.1038/s41586-022-04585-5). In addition, the authors suggest no difference in CAR-T expansion (relma-cel), however, the expansion data in Supp 1b shows that SD/PD patients have rapid expansion of CAR-T on day 1 that the CR/PR patients do not, also agreeing with the data in Supp 1d. All of this seems to suggest that more Teff cells are in the SD/PD patients. Some discussion of this is warranted in the manuscript.

Response: Thank you for raising these issues. A contradictory report has been published demonstrating that the antitumor function of CAR-T is independent of IFN-g production in DLBCL patients ¹, despite the fact that IFN-g is found to be associated with worse therapeutic efficacy of CAR-T.

Regarding the differential expansion rate of CAR-T cells on day 1 between CR/PR patients and SD/PD patients, we noted that the peak value, peak day, and AUC_{Day1-29} did not correlate with the therapeutic response of CAR-T, a phenomenon which was also reported by Ying Z et al ². in which they used the same CAR-T product (Relma-cel) as ours. Thus, we speculated that the short-time expansion of CAR-T is not crucial for the efficacy of Relma-cel. Conversely, the memory T cell expansion and function might be more crucial for the long-lasting effect of CAR-T therapy. For example, in the ZUMA-1 clinical trial, durable responses to Axi-cel were linked to high levels of CCR7+ CD45RA+ T cells in the product, and a greater proportion of T cells with a more juvenile phenotype in the apheresis material was directly associated with a rapid expansion ^{3,4}. In the ZUMA-7 clinical trial, an improved overall survival was linked to a higher proportion of juvenile or stem memory T-cell phenotype cells (CCR7+CD45RA+ T

cells) in the CAR-T product and a lower proportion of differentiated T cells, specifically effector memory cells (CCR7–CD45RA– T cells) ⁵.

Thus, the rapid expansion of CAR-T cells in SD/PD patients could be attributed to higher percentages of effector T cells and lower percentages of memory T cells. In line with this, our results revealed that CAR-T therapy generated a more durable response in CR/PR patients than in SD/PD patients (90 days versus 365 days), a phenomenon consistent with the relatively longer persistence of memory T cells than effector T cells.

Additionally, as our raw data indicated (Figure 1 below), there are two patients (P016 and P103) in the SD/PD group who showed super high copies, leading to a high average value. The large individual variance on day 1 suggested that the long-term effect rather than the early expansion might be more reliable in identifying the therapeutic outcomes of CAR-T therapy. The above discussion has been added in the revised manuscript according to your suggestion (lines 351-357).

Figure 1 Concentration of CAR-T cells in CR/PR and SD/PD patients at early time points after infusion.

References

- [1] Larson, R. C. et al. CAR T cell killing requires the IFN γ pathway in solid but not liquid tumours. *Nature* 604, 563-570, doi:10.1038/s41586-022-04585-5 (2022).
- [2] Ying, Z. et al. Relmacabtagene autoleucel (relma-cel) CD19 CAR-T therapy for adults with heavily pretreated relapsed/refractory large B-cell lymphoma in China. *Cancer Med* 10, 999-1011, doi:10.1002/cam4.3686 (2021).

[3] Jain, M. D. et al. Tumor interferon signaling and suppressive myeloid cells are associated with CAR T-cell failure in large B-cell lymphoma. *Blood* 137, 2621-2633, doi:10.1182/blood.2020007445 (2021).

[4] Biasco, L. et al. Clonal expansion of T memory stem cells determines early anti-leukemic responses and long-term CAR T cell persistence in patients. *Nat Cancer* 2, 629-642, doi:10.1038/s43018-021-00207-7 (2021).

[5] Westin, J. R. et al. Survival with Axicabtagene Ciloleucel in Large B-Cell Lymphoma. *N Engl J Med* 389, 148-157, doi:10.1056/NEJMoa2301665 (2023).

3. Additional details on the way that CRS was graded are needed. Line 112, The median duration of CRS was 10 days on the trial which clashes greatly with other products. Is this because minor issues like fatigue were considered ongoing G1CRS? Or is this due to different management strategies than currently recommended by experts to intervene with tocilizumab even in persistence grade 1 CRS which almost invariably ends the CRS. The low rate of Toci utilization here suggests the latter, but more details and discussion are warranted for the reader to place the clinical results in the context of other trials and products.

Response: It is noteworthy that the median duration in our study (10 days) was numerically longer than other studies using relma-cel (7 days) ^{1,2}. The differences can be attributed to our approach in recording CRS. In this study, investigators recorded the end of CRS following the resolution of fever and other minor symptoms, such as fatigue, headache, and myalgia related to CAR-T therapy. Three patients reported CRS with durations longer than 10 days, contributing to the prolonged median duration (details can be found in Table 1 below).

For your reference, for CRS management in NHL, we adhered to widely accepted approaches, as outlined in the clinical trial protocol. These approaches have been adapted and modified from Lee 2014 ³ and Neelapu SS 2017 ⁴.

In our study, intensive monitoring, follow-up, and supportive treatments were employed for patients with mild CRS symptoms. For non-life-threatening grade 1 events, tocilizumab was not administered, as evidenced by the relatively lower rate of tocilizumab use. The manuscript has been updated accordingly, and further details can be found in lines 340-350.

Further comprehensive information can be found at “CRS Grading Criteria (Lee 2014)” and “Approaches to CRS Management in NHL” in the Clinical Trial Protocol on pages 43-44.

Table 1: Details for the 3 patients underwent CRS.

Patient ID	Onset of CRS (days after infusion)	End of CRS (days after infusion)	Duration of CRS (days after infusion)	The highest grade of CRS	Reason for longer duration of CRS
P003	0	17	17	2	Constant fatigue and headache
P005	7	20	13	1	Constant fatigue and myalgias
P006	4	16	12	2	Constant fatigue and headache

References

- [1] Yan, Z.X., L. Li, W. Wang, B.S. OuYang, S. Cheng, L. Wang, et al., Clinical Efficacy and Tumor Microenvironment Influence in a Dose-Escalation Study of Anti-CD19 Chimeric Antigen Receptor T Cells in Refractory B-Cell Non-Hodgkin's Lymphoma. *Clin Cancer Res*, 2019. 25(23): p. 6995-7003.
- [2] Ying Z, Y.H., Guo Y, Li W, Zou D, Zhou D, Wang Z, Zhang M, Wu J, Liu H, Zhang P, Yang S, Zhou Z, Zheng H, Song Y, Zhu J. , Relmacabtagene autoleucel (relma-cel) CD19 CAR-T therapy for adults with heavily pretreated relapsed/refractory large B-cell lymphoma in China. *Cancer Med*, 2021. 10(3): p. 999-1011.
- [3] Lee DW, Gardner R, Porter DL, et al. Current concepts in the diagnosis and management of cytokine release syndrome. *Blood* 2014;124(2):188-195
- [4] Neelapu, et al. Chimeric antigen receptor T-cell therapy--assessment and management of toxicities. *Nat Rev Clin Oncol* 2017;148

4. The manuscripts states that patient 2 was excluded from efficacy analysis because they developed Hodgkin's lymphoma 13 months after treatment. The authors should not exclude this patient from efficacy analysis and instead censure the patient from the PFS endpoint at 13 months. Also please confirm they were included in the safety data. If the patient was actually not eligible because they had hodgkins before CAR-T, that should be disclosed.

Response: We sincerely value your diligence regarding this matter. Here, we confirmed that patient 2 has indeed been included in the safety dataset (12 patients).

We excluded patient 2 from the efficacy dataset (11 patients) as we believe that the patient probably had a second disease which is Hodgkin's lymphoma (HL), which shares similar clinical manifestations with DLBCL, for example, fever, fatigue, weight loss, and lymph node enlargement. Thus, the diagnosis or distinction of these two diseases heavily depended on the pathologist's experience, that is, while they are examining cancer cells under a microscope.

Such combined cases are referred to as composite lymphoma (CL), which indicates more than one histological variety of lymphoma developing in a single patient. In the literature, more than 20 combined cases of HL+DLBCL were reported ¹.

According to a retrospective evaluation of our study, we assumed that patient 2 was likely having both DLBCL and HL prior to the study enrollment. At the first diagnosis during enrollment, employing right cervical lymph node biopsy only confirmed DLBCL (CD20 + and CD30+), and then 13 months later, the patient was diagnosed as HL through left cervical lymph node biopsy (CD30+).

The assessment of CAR-T therapy efficacy in HL and DLBCL is quite similar, as both typically present with areas of elevated glucose uptake on PET-CT scans. However, given the potential influence of HL on the evaluation of CAR-T therapy efficacy in DLBCL, we decided, after in-depth discussions, to exclude patient 2 from the efficacy analyses. This was done to prevent the introduction of confounding factors associated with the presence of a second disease, that is, HL, into the efficacy assessments.

References

[1] Wang, J. & Zhang, R. Composite lymphoma of cervical lymph nodes with classical Hodgkin lymphoma and diffuse large B cell lymphoma: a case report and literature review. *Ann Palliat Med* **9**, 3651-3662, doi:10.21037/apm-20-1290 (2020).

5. Line 118, a patient is described as dying from “severe abdominal infection”. This needs more detail. Was this enterocolitis or typhlitis or abdominal abscess or ?????

Response: We greatly appreciate your observation and have provided the detailed description in the revised manuscript. At diagnosis, the patient suffered from a mass of 7 cm, which infiltrated the surround organ, involving intestines, adrenal glands, uterus, and adnexa. After CAR-T therapy, the patient developed a renal abscess secondary to a *Listeria* infection, which was further complicated by *Pseudomonas aeruginosa* septicemia.

Eligibility during enrollment was assessed and the eligibility was carefully re-assessed when the CAR-T product had been manufactured. Following lymphodepletion chemotherapy, the patient received JW029CAR-T cell infusion on November 21, 2020. Gratefully, the patient achieved complete remission (CR) on day 28 and day 90. Unfortunately, about 5 months following the successful remission, the patient experienced intermittent fever. An enhanced CT scan observed an abnormal density shadow in the left kidney. Subsequent pathogenic microorganism NGS testing of the left kidney mass puncture indicated an infection caused by

Listeria monocytogenes. Furthermore, pathogenic microorganism NGS of the peripheral blood confirmed the presence of Pseudomonas aeruginosa infection. The patient received targeted antibiotic therapy including ampicillin, beta-lactams, and quinolones, to treat the identified infections. Despite the above extensive efforts, the patient unfortunately succumbed to sepsis. The manuscript has been revised accordingly. See lines 124-126.

6. The importance of M2 macrophages in the tumor prior to CAR-T has been established although other groups suggest an association of M2 in tumor with both systemic/tumor inflammation and circulating suppressive MDSCs (Jain et al, Blood 2021, DOI:10.1182/blood.2020007445). Can the authors provide data on the systemic or tumor inflammation in relation to M2 macrophages in these patients? In addition, the impact of the negative impact of high CRP and ferritin is well described. Regardless, a comment in the discussion about the link between tumor, inflammation, and myeloid cells in DLBCL resistance to CAR-T is warranted.

Response: Thank you for your suggestion. As shown in Figure 1 below (Figure 2d in our manuscript), we noted that SD/PD patients showed higher percentages of M2 macrophages than CR/PR patients in the tumor microenvironment prior to the CAR-T therapy, which is in line with previous reports¹. Surprisingly, apart from M2 macrophages, the percentages of multiple myeloid subpopulations, including cDC1, cDC2, pDC, M1 macrophages, and M2 macrophages, were higher in the SD/PD patients, with M2 macrophages as the most pronounced subset. Conversely, in our study, the CRP and ferritin levels were comparable between SD/PD and CR/PR patients prior to preinfusion (Figure 2 below). This inconsistency might be that ferritin and CRP levels are affected by multiple factors, such as iron metabolism status or the inflammatory status of cancer patients. We have addressed these issues in the discussion section (lines 377-380).

For these patients, we have already provided data related to inflammation and M2 macrophages in Table 1 below.

Figure 1 The percentage of cDC1, cDC2, cDC3, pDC, M1 macrophage, M2 macrophage subclusters in CR and PD patients before CAR-T cell therapy.

Figure 2. The systemic inflammation-related cytokine signatures before infusion

NO.	inflammation at Day 0				inflammation Peak value				inflammation Peak day				Percentage of M2 in myeloid cells before CAR-T therapy
	CRP mg/L	Ferritin ng/mL	IL-6 pg/mL	TNF- α pg/mL	CRP mg/L	Ferritin ng/mL	IL-6 pg/mL	TNF- α pg/mL	day	day	day	day	
P003	86.2	1193.2	11.2	2.4	127.3	1062	1830	2.4	6	1	7	1	0.47%
P005	6.4	104.7	6.3	2.4	33.8	92.7	21.3	5.6	11	1	12	1	18.95%
P006	14.9	306.7	4.1	2.4	85.8	1356	3598	2.4	6	9	7	1	/
P007	3.1	2.4	6.3	2.4	5.6	2.4	11.4	2.6	3	1	7	8	8.62%
P008	8.9	823.8	2.5	2.4	6.7	1247	7	2.4	14	2	12	1	/
P009	17.1	763.1	25.1	2.4	28.2	801.2	34.4	2.4	6	29	3	1	/
P010	21.5	338.9	15.5	2.4	65.1	678.9	504	2.4	6	29	29	1	1.82%
P012	54.7	1957.5	8.3	2.4	45.6	1036	33.8	2.4	6	6	5	2.4	7.24%
P016	53.8	34.5	98	2.4	41.7	181.6	104.6	8.2	2	29	29	8	/

Table 1. Inflammation and M2 macrophages in the table.

References

[1] Jain, M. D. et al. Tumor interferon signaling and suppressive myeloid cells are associated with CAR T-cell failure in large B-cell lymphoma. *Blood* 137, 2621-2633, doi:10.1182/blood.2020007445 (2021)

7. The description of results on mutational analysis in Supp Fig 4 seems cursory. Recent reports outline the importance of tumor mutations in the efficacy of CAR-T (Sworder et al, *Cancer Cell*, 2023, DOI:10.1182/bloodadvances.2020002394; and Jain et al, *Blood* 2022, doi:10.1182/blood.2021015008). It is unclear if this data is necessary or if it is robust given the small numbers here.

Response: We appreciate and total agree with your suggestion. We surmise that including tumor mutation data was unnecessary for the following reasons: (1) In the present study, we highlighted on the tumor immune microenvironment of CAR-T-treated patients, with minimal attention on tumor mutations. (2) As you emphasized, the analysis of tumor mutation typically requires a relatively large population compared to immune profiling of patients. (3) Gene mutation regulates cell behaviors through changing protein functions, which can also be influenced by multiple mechanisms, such as gene expression (transcriptional regulation), translation, or post-translational modifications. The relative contribution of each mechanism to protein function regulation is complicated and thus is beyond the scope of our present work. Thus, for the main conclusion of our work, the mutational analysis in the original manuscript is not necessary. The reference to tumor mutations has been removed from the original article. In the future, we shall conduct further investigation on the relationship between prognosis and mutation in a larger clinical trial in this population.

8. It is a bit unclear based upon the results whether the decreased efficacy associated with M2 macrophages is associated with numbers (more M2 for PD/SD patients) or if it is due to differences in M2 macrophages in these patients as suggested online 175. Could the authors discuss their beliefs on this within the discussion?

Response: In the present work, a pronounced enhancement of M2 macrophage functions in PD/SD patients was observed, as evidenced not only by the increase in the percentages of M2 macrophages, but through the enrichment of the “cholesterol efflux” pathway in M2 macrophages. Both changes are regarded as negative regulators in the therapeutic efficacy of CAR-T treatment. For example, our previous work indicated that both the number and function of M2 macrophage affected the therapeutic response of CAR-T therapy^{1,2}. Additionally,

membrane-cholesterol efflux reportedly facilitates TAM-mediated tumor progression^{3,4}. Thus, we believe that the decreased CAR-T efficacy is associated with alternations in the number (lines 165-167) and function (lines 181-186) of M2 macrophages. We have addressed these issues in the discussion section (lines 396-402).

References

- [1] Yan, Z. X. et al. Clinical Efficacy and Tumor Microenvironment Influence in a Dose Escalation Study of Anti-CD19 Chimeric Antigen Receptor T Cells in Refractory B-Cell Non Hodgkin's Lymphoma. *Clin Cancer Res* 25, 6995-7003, doi:10.1158/1078-0432.CCR-19-0101 (2019).
- [2] Yan, Z. et al. Immunosuppressive tumor microenvironment contributes to tumor progression in diffuse large B-cell lymphoma upon anti-CD19 chimeric antigen receptor T therapy. *Front Med*, doi:10.1007/s11684-022-0972-8 (2023).
- [3] Goossens P, Rodriguez-Vita J, Etzerodt A, Masse M, Rastoin O, Gouirand V, et al. Membrane Cholesterol Efflux Drives Tumor-Associated Macrophage Reprogramming and Tumor Progression. *Cell Metab*. 2019;29(6):1376-89 e4.
- [4] Nelson ER, Wardell SE, Jasper JS, Park S, Suchindran S, Howe MK, et al. 27 Hydroxycholesterol 831 links hypercholesterolemia and breast cancer pathophysiology. *Science*. 2013;342(6162):1094-8

9. It would be helpful if Figure 2 included a small table outlining which of the patients included in Single cell analysis were PD/SD or CR.

Response: Thank you for your constructive comments. A detailed data on the efficacy for each single-cell patient have been added in Figure 2a.

10. Single cell analysis of cell numbers/type can be skewed if all or most of a certain type of cell come from 1 or only a few samples. Can the authors address this concern as it relates to results in Figure 2d and Figure 3a.

Response: We sincerely appreciate the reviewer's meticulous review for the potential bias in our single-cell analysis results. In response to this concern, in our study, we employed measures to ensure a minimum of two samples at each time point or grouping. Specifically, for Figure 2d, two samples were incorporated from patients who achieved complete responses (CR), and three samples from patients with progressive disease (PD) were also included.

In Figure 3a, the sample distribution is structured as follows: before CAR-T cell therapy (n=5), at CAR-T cell expansion (n=3), and at the disease progression stage (n=2). It is noteworthy that the limitation of a restricted sample size is a common challenge encountered in CAR-T studies, particularly within clinical trials. Despite this constraint, concerted efforts were exerted to maximize patient inclusion in our study. While we recognize the benefits of larger sample sizes, our current dataset offers meaningful and representative insights.

As we proceed with our research, in future endeavors, we are fully committed to expand our sample size, thereby further enhancing the robustness and reliability of our findings.

11. The CellChat data in Figure 5 is interesting and adds value, however the methods are unclear. Were these results from your single cell data, but calibrated on Breast Cancer data? Or are these results from a Breast Cancer data set? If the latter this is less interesting. If the former, did the authors develop this methodology or can citations be used to validate it?

Response: The single-cell sequencing data utilized in the analysis conducted using CellChat originated from our clinical trial. This dataset comprised samples collected at distinct time points, including before CAR-T cell therapy (n=5), during CAR-T cell expansion (n=3), and at the disease progression (n=2). Among these samples, two were obtained from patients who achieved complete responses (CR), while three were collected from patients who exhibited progressive disease (PD) prior to CAR-T cell therapy. ScRNA-seq data preprocessing and Cell-cell ligand-receptor communication analysis have been amended at lines 528-539 and 635-646.

The analytical methodology employed in the CellChat analysis drew inspiration from a previously published article that addressed T cell-B cell crosstalk in the context of triple-negative breast cancer ¹.

References

[1] Ding, S. et al. Single-cell atlas reveals a distinct immune profile fostered by T cell-B cell crosstalk in triple negative breast cancer. *Cancer Commun (Lond)* 43, 661-684, doi:10.1002/cac2.12429 (2023).

12. In some of the downstream analysis of single cell results it is unclear if the data was derived from all cells, or only M2 or T cell subsets. Specifically, Figure 6i shows ER stress

pathways across time, but the impact of ER stress is different in myeloid cells and T cells, so it would be important to clarify.

Response: Thank you for your question, and our apologies for any confusion this has caused. In the recently revised manuscript, comprehensive annotations specifying the origin of the sample have been included, encompassing the overall cell population as well as M2 and T cells. Figure 3 includes the analysis of M2, and Figures 4 and 6 include the analysis of the T cells. We have specified the corresponding cell groups again in the figure legend. Figure 6i includes the functional analysis of CD8 Teff cells, which is elucidated in both figure legend and figure at lines 1081.

Reviewer #3 (expert in cholesterol metabolism):

In this study by Zi-Xun Yan et al., the authors revealed that the cholesterol efflux of M2 macrophage played a key role in CAR-T cell therapy resistance in primary refractory DLBCL by triggering CD8+ T-cell exhaustion. The authors performed scRNA-seq on 10 fresh biopsy tissues from 5 primary refractory DLBCL patients, including 5 samples before CAR-T cell therapy, 3 samples at CAR-T cell expansion and 2 samples at disease progression and found that cholesterol efflux pathways were significantly upregulated at disease progression stage. ABCA1 (membrane cholesterol efflux transporters) knockdown rescued the immune-suppression function mediated by M2 macrophages. Overall, the study appears to be well designed and data presented are all in good quality too. Given that a major portion of the data were obtained from clinical trial specimens, the findings will likely be of great interests to the field of cancer immunity and cancer immune therapy. The following few comments can be considered for its revision.

1) To enhance the significance of the study and its findings, it is highly recommended that more samples from additional patients, particularly from the PD patients, are examined. Currently, only two such samples are included in the study.

Response: I extend my sincere gratitude to the reviewer for their thorough evaluation and insightful recommendations, which have significantly enriched our manuscript.

Our study represents an exploratory phase I clinical trial, focusing primarily on safety assessment. Given the uncertainties surrounding the efficacy of Relma-cel CAR-T therapy in first-line, primary refractory DLBCL, ethical considerations lead ethics committees to impose sample size limitations in phase I trials, prioritizing patient welfare. This restriction aims to facilitate preliminary observations and efficacy assessments while mitigating potential risks. Eleven patients underwent efficacy assessment at 3 month, five patients had disease progression, Tumor tissue puncture was successfully performed in three patients before CAR-T therapy and during CAR-T cell expansion, with challenges faced during disease progression due to anatomical location or patient preferences, limiting punctures to two progressing patients. Recognition of the key role of cholesterol efflux in M2 macrophage-mediated CAR-T therapy resistance in primary refractory patients emerged at various treatment phases in dynamically observed patients. We acknowledge the current sample size limitation and commit to validate our findings in a more extensive cohort.

2). The current manuscript seems lacking figure legends, which makes it difficult to assess some of the data just by reading the Results and M&M.

Response: We are sorry for the inconvenience. We still do not know why Reviewer 3 can not find the figure legends. The updated manuscript with the legends has been re-uploaded.

3). In Figure 2f, the authors claimed that there was no significant difference in T cells between complete response (CR) and progressive disease (PD) patients. However, the data presented appear to show that the proportions of T cell subtypes did show some significant differences in CR and PD patients. The authors may need to perform either additional experiments or additional analyses of the individual T cell subtypes for statistical significance.

Response: We sincerely appreciate your valuable suggestion. We recognize your keen observation that the data may suggest variations in the proportions of T-cell subtypes between patients with complete responses (CR) and those with progressive disease (PD). It's worth noting that our study included two CR patients and three PD patients before CAR-T therapy. Following your suggestion, we conducted a Wilcoxon rank-sum test, which, within our current dataset, did not reveal statistically significant differences among the T-cell subtypes (Table 1 below). We acknowledged that the limited sample size and inherent variability in clinical patient samples could influence the reliability of statistical power. To provide clarity, we have included the percentages corresponding to each patient's data (Table 2 below). In our forthcoming research, we are dedicated to expanding our sample size to enhance the statistical reliability.

Clusters	group1	group2	n1	n2	statistic	p-value	method
NK	CR	PD	2	3	2	0.8	wilcox.test
CD8_Teff	CR	PD	2	3	3	1.0	wilcox.test
CD8_Tex_1	CR	PD	2	3	1	0.4	wilcox.test
CD8_Tex_2	CR	PD	2	3	3	1.0	wilcox.test
CD8_Tex_3	CR	PD	2	3	5	0.4	wilcox.test
CD8_Tex_4	CR	PD	2	3	5	0.4	wilcox.test
CD8_Tex_5	CR	PD	2	3	4	0.8	wilcox.test
CD8_Tex_6	CR	PD	2	3	6	0.2	wilcox.test
CD8_Prolif_1	CR	PD	2	3	5	0.4	wilcox.test
CD8_Prolif_2	CR	PD	2	3	6	0.2	wilcox.test
CD8_Prolif_3	CR	PD	2	3	1	0.4	wilcox.test

CD4_Naive	CR	PD	2	3	6	0.2	wilcox.test
CD4_Treg	CR	PD	2	3	4	0.8	wilcox.test
CD4_Mem	CR	PD	2	3	2	0.8	wilcox.test
CD4_Th1_like	CR	PD	2	3	4	0.8	wilcox.test
LowQual	CR	PD	2	3	0	0.2	wilcox.test

Table 1 Wilcoxon rank-sum test among the T cell subtypes between CR and PD patients

Clusters	Absolute cell number					Relative percentage (%)				
	CR	PD	CR	PD	CR	PD	CR	PD	CR	PD
NK	202	130	9	114	135	2.8	4.2	3.1	3.4	18.9
CD8_Teff	285	148	12	128	68	4.0	4.7	4.2	3.8	9.5
CD8_Tex_1	316	167	13	213	56	4.4	5.3	4.5	6.3	7.9
CD8_Tex_2	616	400	21	410	104	8.6	12.8	7.3	12.1	14.6
CD8_Tex_3	205	358	6	79	27	2.9	11.4	2.1	2.3	3.8
CD8_Tex_4	507	362	6	238	65	7.1	11.6	2.1	7.1	9.1
CD8_Tex_5	750	109	7	121	30	10.4	3.5	2.4	3.6	4.2
CD8_Tex_6	1325	685	28	592	98	18.5	21.9	9.8	17.5	13.7
CD8_Prolif_1	105	91	2	34	13	1.5	2.9	0.7	1.0	1.8
CD8_Prolif_2	159	100	3	74	13	2.2	3.2	1.0	2.2	1.8
CD8_Prolif_3	22	37	15	74	6	0.3	1.2	5.2	2.2	0.8
CD4_Naive	568	129	5	139	19	7.9	4.1	1.7	4.1	2.7
CD4_Treg	1284	147	22	259	18	17.9	4.7	7.7	7.7	2.5
CD4_Mem	631	162	27	678	28	8.8	5.2	9.4	20.1	3.9
CD4_Th1_like	75	26	1	52	1	1.0	0.8	0.3	1.5	0.1
LowQual	131	79	109	170	32	1.8	2.5	38.1	5.0	4.5

Table 2 The percentages of T cell subtypes in each patient

4) In Figure 2g, the authors presented the pathways significantly enriched in M2 macrophages in CR and PD patients. It is better to use -logP value to sort these pathways rather than Gene Ratio. Also, changing the bar color from currently blue to something lighter will display the specific pathways/programs better.

Response: We sincerely appreciate the reviewer's constructive feedback regarding the presentation of Figure 2g. To enhance the clarity of the figure, we have implemented the following improvements: Firstly, we have reorganized the pathways based on -log₁₀ adjusted P-values, which provides a more informative ranking. Secondly, we have adjusted the bar color to a lighter shade, ensuring better visibility of specific pathways. The updated figure is now available in our revised manuscript for your reference.

5) In Figure 3 and 6, the authors tested the effects of M2 macrophages with ABCA1 gene knockdown that resulted in inhibited cholesterol efflux in M2 macrophages. The authors should consider to provide the evidence that activation of the cholesterol efflux can further enhance the immunosuppression effects of M2 cells by genetic (e.g. overexpression ABCA1) or using pharmacological approaches.

Response: We thank the reviewer for your helpful comment. To validate the impact of cholesterol efflux on the immunosuppressive properties of M2 macrophages, we utilized 9cRA¹ (pharmacological approach) to promote cholesterol efflux in M2 macrophages (Supplementary Fig. 4a). As expected, a significant decrease in total cholesterol content in the co-culture system was observed after 9cRA treatment, accompanied by an increase in cholesterol content in the culture medium (Supplementary Fig. 4b). Notably, macrophages exhibited significant polarization towards the M2 phenotype as assessed by CD206 expression (Supplementary Fig. 4c). In CAR-T cell cytotoxicity experiments, it was observed that the cytotoxicity of CAR-T cells in the M2^{9cRA}-CAR19 co-culture group was significantly decreased when compared to control group (Supplementary Fig. 4d). These findings collectively suggest that the increased cholesterol efflux in M2 macrophages inhibits the anti-tumor function of CAR-T cells. Please refer to lines 223-234 and 1127-1139 for the revision.

Supplementary figure 4

Supplementary Figure 4. Increased cholesterol efflux from M2 macrophages using 9cRA pharmacological approaches enhance the immunosuppression effects.

(a) In vitro, coculture models were established to evaluate the effects of M2 macrophages and M2 macrophages with 9cRA on CAR-T cell cytotoxicity against CD19-expressing DLBCL cells (DB cells). (b) The impact of CAR19 cells on the total or secreted cholesterol levels of M2 macrophages was evaluated. (c) The expression of CD206 on macrophages in the indicated groups was evaluated through flow cytometry. (d) The cytotoxic effect of CAR19 cells on

CD19-expressing DLBCL cells (DB) and CD19-nonexpressing acute promyelocytic leukemia cells (HL60) was assessed using a luciferase-based CTL assay. Data are presented as mean \pm s.e.m. Statistical analysis was performed using two-way ANOVA with Tukey's multiple comparison tests.

References

[1] Goossens, P. et al. Membrane Cholesterol Efflux Drives Tumor-Associated Macrophage Reprogramming and Tumor Progression. *Cell metabolism* 29, 1376-1389.e1374, doi:10.1016/j.cmet.2019.02.016 (2019).

6) The authors used M2 macrophage-PBMC co-culture system with just one cell line to test their hypothesis that cholesterol efflux of M2 macrophages that triggered immunosuppression that caused CAR-T cell therapy resistance in vitro. Data from co-culture with cells isolated from the clinic or other sources will enhance the significance of the study. If possible, the authors may consider to establish or use humanized mouse models for obtaining evidence to support their major findings.

Response: Thanks again for your constructive suggestions. According to your comments, we co-cultured tumor cells and PBMC-derived M2 macrophages¹ from DLBCL patients to mimic the TME. The cholesterol levels in cells and culture supernatant, the percentages of T-cells, and levels of immune checkpoints were quantified (Supplementary Fig. 6a). The M2-PBMC group exhibited a significantly lower cholesterol level than the control group and PBMC alone group. In contrast, a higher cholesterol level was detected in the culture supernatant of M2-PBMC group. Importantly, ABCA1 inhibition in macrophages (M2^{shABCA1} PBMC group) restored normal cholesterol levels in the M2-PBMC co-culture system (Supplementary Fig. 6b). Furthermore, a significant reduction in the percentage of CD8⁺ T-cells (Supplementary Fig. 6c and d), and an increase in the percentages of PD1⁺ (Supplementary Fig. 6e and f) and LAG3⁺ T-cells (Supplementary Fig. 6g and h) were detected in the M2-PBMC group, which were reversed by macrophage ABCA1 knockdown, suggesting that macrophage cholesterol efflux induces T cell exhaustion. Please refer to lines 318-322, 684-690 and 1169-1179 for the revision.

Supplementary Figure 6. T-cell exhaustion induced by cholesterol efflux from PBMC-derived M2 macrophages.

(a) Flowchart of the *in vitro* assay. PBMCs were cocultured with control, M2 macrophages, and ABCA1-knockdown M2 macrophages (M2^{shABCA1}) for 48 h, in the presence of DB cells to mimic the TME. Cholesterol levels in cells and culture supernatant, the percentages of T-cells, and immune checkpoint genes were quantified. (b) Quantification of total cholesterol levels in cells and medium. Flow cytometry analysis of the percentages of CD8⁺ T-cells (c, d), PD1⁺ cells (e, f), and LAG3⁺ T-cells (g, h).

References

[1] Goossens, P. et al. Membrane Cholesterol Efflux Drives Tumor-Associated Macrophage Reprogramming and Tumor Progression. *Cell metabolism* 29, 1376-1389.e1374, doi:10.1016/j.cmet.2019.02.016 (2019).

7). The writing of this current manuscript needs to be significantly improved for its clarity.

The followings are just some sentences that need better clarity:

The sentence in line 38 – 40, too vague; it can be expanded with additional description.

Line 81 and 120 needs additional words to give some details.

There are several long, run-on sentences: on line 200 – 207 and elsewhere.

Response: We appreciate your insightful suggestion and have revised the manuscript by a native speaker.

REVIEWERS' COMMENTS

Reviewer #2 (Remarks to the Author):

Thank you for addressing my comments.

Reviewer #3 (Remarks to the Author):

The authors have addressed the comments of this reviewer satisfactorily mostly, except that the following information needs to be provided.

- 1). Please spell out 9cRA in the Results and Figure legend and point out in the Results that 9cRA has been previously used to promote cholesterol efflux in
- 2). Please include the information on the use of 9cRA in the Methods.
- 3). Statistics information is missing for the lower right panel showing the the % of lysis of Supplementary Figure 4.
- 4). For Supplementary Figure 6, please provide information, in the Methods, on the "Control" in the four different PBMC samples. Also, please provide information on the source(s) the PBMCs isolated, such as the individual people (normal or with disease, age, sex,).
- 5). Please provide WB data showing the ABCA1 KD efficiency and the statistics method(s).

REVIEWER COMMENTS

Reviewer #1 (expert in biostatistics):

Paper Review:

Summary:

The paper presents a Phase I trial assessing the efficacy and safety of CAR-T cell therapy in primary refractory DLBCL patients. The authors assert that CAR-T cell therapy shows promise in enhancing the anti-tumor effect in DLBCL patients.

Major Comments:

A notable concern in this study is the absence of support for these claims through power analyses or sample size calculations within the paper. The final sample size employed for drawing these conclusions comprises only 11-12 patients, which appears smaller than the customary sample size of approximately 20. It would greatly benefit the study to provide insights into how the power analysis was conducted during the study's design phase.

Minor Comments:

There are a few additional minor points to address:

- Lines 706-708: The statement made here is somewhat perplexing. It appears that all statistical tests were carried out under the assumption of a normal distribution. Is this perhaps a typographical error?

- Another minor issue pertains to the selection of certain algorithm parameters, such as those for DDRTree and clustering. It would be helpful to clarify how these algorithm parameters were determined and fixed. This would help ensure reproducibility of the work.

- Is the code used for the analyses open-source? A link would be helpful.

Reviewer #2 (expert in haematological malignancies):

Thank you for the opportunity to review the manuscript by Yan et al entitled "Cholesterol efflux of M2 macrophage in CAR-1 T cell therapy resistance: A phase I study of primary refractory DLBCL (JWCAR029-003)". I enjoyed reading the story and believe it adds new and interesting information to the conversation about tumor associated features mediating resistance to CAR-T for DLBCL. There are some grammatical features that require attention, however the message is clear: M2 mediated cholesterol efflux plays a role in T cell function within the DLBCL environment and is associated with response in this trial. I have the following comments which I believe should be addressed:

1. The clinical trial itself is interesting in that only Primary Refractory patients were enrolled. Was this data reported in another peer reviewed manuscript? If yes, it should be cited. If no, I believe the significance of the trial being for only those with Primary Refractory disease should be highlighted in the introduction as this nuance otherwise is not highlighted until the discussion where the authors correctly point out that patients with refractory disease fare with CAR-T than those with relapsed disease. Unless this nuance is clear the reader could incorrectly assume this CAR-T is inferior (based upon CR/ORR rates on line 97) as compared to liso-cel or axi-cel, where instead a more aggressive population was treated here.

2. The finding presented in Supplemental figure 1d is interesting and agrees with ZUMA-1 where more IFN-g release upon in vitro stim of the product was associated with worse efficacy (Locke et al, Blood Advances, 2020, DOI:10.1182/bloodadvances.2020002394) and data demonstrating that

IFN-g from CAR-T cells is dispensable for effectiveness in hematologic malignancies (Larson et al, Nature, 2022, DOI:10.1038/s41586-022-04585-5). In addition, the authors suggest no difference in CAR-T expansion, however the expansion data in Supp 1b showing that SD/PD patients have rapid expansion of CAR-T on day 1 that the CR/PR patients do not, also agreeing with the data in Supp 1d. All of this seems to suggest that more Teff cells are in the SD/PD patients. Some discussion of this is warranted in the manuscript.

3. Additional details on the way that CRS was graded are needed. Line 112, The median duration of CRS was 10 days on the trial which clashes greatly with other products. Is this because minor issues like fatigue were considered ongoing G1CRS? Or is this due to different management strategies than currently recommended by experts to intervene with tocilizumab even in persistence grade 1 CRS which almost invariably ends the CRS. The low rate of Toci utilization here suggests the latter, but more details and discussion are warranted for the reader to place the clinical results in context of other trials and products.

4. The manuscript states that patient 2 was excluded from efficacy analysis because they developed Hodgkin's lymphoma 13 months after treatment. The authors should not exclude this patient from efficacy analysis and instead censure the patient from PFS endpoint at 13 months. Also please confirm they were included in the safety data. If the patient was actually not eligible because they had Hodgkin's before CAR-T, that should be disclosed.

5. Line 118, a patient is described as dying from "severe abdominal infection". This needs more detail. Was this enterocolitis or typhlitis or abdominal abscess or ?????

6. The importance of M2 macrophages in the tumor prior to CAR-T has been established although other groups suggest an association of M2 in tumor with both systemic/tumor inflammation and circulating suppressive MDSCs (Jain et al, Blood 2021, DOI:10.1182/blood.2020007445). Can the authors provide data on the systemic or tumor inflammation in relation to M2 macrophages in these patients? In addition, the impact of the negative impact of high CRP and ferritin is well described. Regardless, a comment in the discussion about the link between tumor, inflammation, and myeloid cells in DLBCL resistance to CAR-T is warranted.

7. The description of results on mutational analysis in Supp Fig 4 seems cursory. Recent reports outline the importance of tumor mutations in efficacy of CAR-T (Sworder et al, Cancer Cell, 2023, DOI:10.1182/bloodadvances.2020002394 ; and Jain et al, Blood 2022, doi:10.1182/blood.2021015008). It is unclear if this data is necessary or if it is robust given the small numbers here.

8. It is a bit unclear based upon the results whether the decreased efficacy associated with M2 macrophages is associated with numbers (more M2 for PD/SD patients) or if it is due to differences in M2 macrophages in these patients as suggested on line 175. Could the authors discuss their beliefs on this within the discussion?

9. It would be helpful if Figure 2 included a small table outlining which of the patients included in Single cell analysis were PD/SD or CR.

10. Single cell analysis of cell numbers/type can be skewed if all or most of a certain type of cell come from 1 or only a few samples. Can the authors address this concern as it relates to results in Figure 2d and Figure 3a.

11. The CellChat data in Figure 5 is interesting and adds value, however the methods are unclear. Were these results from your single cell data, but calibrated on Breast Cancer data? Or are these results from a Breast Cancer data set? If the latter this is less interesting. If the former, did the authors develop this methodology or can citations be used to validate it?

12. In some of the downstream analysis of single cell results it is unclear if the data was derived from all cells, or only M2 or T cell subsets. Specifically, Figure 6i shows ER stress pathways across time, but the impact of ER stress is different in myeloid cells and T cells, so it would be important to clarify.

Reviewer #3 (expert in cholesterol metabolism):

In this study by Zi-Xun Yan et al., the authors revealed that the cholesterol efflux of M2 macrophage played a key role in CAR-T cell therapy resistance in primary refractory DLBCL by triggering CD8+ T-cell exhaustion. The authors performed scRNA-seq on 10 fresh biopsy tissues from 5 primary refractory DLBCL patients, including 5 samples before CAR-T cell therapy, 3 samples at CAR-T cell expansion and 2 samples at disease progression and found that cholesterol efflux pathways were significantly upregulated at disease progression stage. ABCA1 (membrane cholesterol efflux transporters) knockdown rescued the immune-suppression function mediated by M2 macrophages. Overall, the study appears to be well designed and data presented are all in good quality too. Given that a major portion of the data were obtained from clinical trial specimens, the findings will likely be of great interests to the field of cancer immunity and cancer immune therapy. The following few comments can be considered for its revision.

- 1) To enhance the significance of the study and its findings, it is highly recommended that more samples from additional patients, particularly from the PD patients, are examined. Currently, only two such samples are included in the study.
- 2). The current manuscript seems lacking figure legends, which makes it difficult to assess some of the data just by reading the Results and M&M.
- 3). In Figure 2f, the authors claimed that there was no significant difference in T cells between complete response (CR) and progressive disease (PD) patients. However, the data presented appear to show that the proportions of T cell subtypes did show some significant differences in CR and PD patients. The authors may need to perform either additional experiments or additional analyses of the individual T cell subtypes for statistical significance.
- 4) In Figure 2g, the authors presented the pathways significantly enriched in M2 macrophages in CR and PD patients. It is better to use $-\log P$ value to sort these pathways rather than Gene Ratio. Also, changing the bar color from currently blue to something lighter will display the specific pathways/programs better.
- 5) In Figure 3 and 6, the authors tested the effects of M2 macrophages with ABCA1 gene knockdown that resulted in inhibited cholesterol efflux in M2 macrophages. The authors should consider to provide the evidence that activation of the cholesterol efflux can further enhance the immunosuppression effects of M2 cells by genetic (e.g. overexpression ABCA1) or using pharmacological approaches.
- 6) The authors used M2 macrophage-PBMC co-culture system with just one cell line to test their hypothesis that cholesterol efflux of M2 macrophages that triggered immunosuppression that caused CAR-T cell therapy resistance in vitro. Data from co-culture with cells isolated from the clinic or other sources will enhance the significance of the study. If possible, the authors may consider to establish or use humanized mouse models for obtaining evidence to support their major findings.
- 7). The writing of this current manuscript needs to be significantly improved for its clarity. The followings are just some sentences that need better clarity:
The sentence in line 38 – 40, too vague; it can be expanded with additional description.
Line 81 and 120 needs additional words to give some details.
There are several long, run-on sentences: on line 200 – 207 and elsewhere.